# Difficulty-Aware Reasoning for Mobile GUI Automation via Reinforcement Fine-Tuning

## Abstract

Automating GUI tasks remains challenging due to layout complexity, element density, and intent ambiguity, which requires effective and efficient reasoning to facilitate each operation. Existing agents typically employ a uniform chain-of-thought (CoT) reasoning process for all actions, a one-size-fits-all approach that incurs unnecessary computational overhead and even performance degradation on trivial steps. To address this, we introduce **AdaGUI-R1**, a GUI agent that pioneers a difficulty-aware reasoning paradigm by dynamically modulating its reasoning depth based on action complexity. Our methodology consists of reasoning inducing and reasoning enhancing. During reasoning inducing, we introduce a self-supervised mechanism to generate high-quality, difficulty-aware reasoning trajectories. Fine-tuning on this curated data endows the agent with the fundamental capability to adjust its reasoning depth according to action complexity. Subsequently, Group Adaptive Policy Optimization (GAPO) algorithm is implemented to enhance reasoning performance. It leverages an adaptive thought reward to encourage thinking on challenging steps, and a novel exploration reward with a difficulty-aware Gaussian bandwidth to improve action accuracy. Extensive experiments demonstrate that AdaGUI-R1 sets a new state-of-the-art. It concurrently reduces unnecessary reasoning tokens by 40% while improving action accuracy by 5%, underscoring the power of adaptive reasoning in GUI automation.

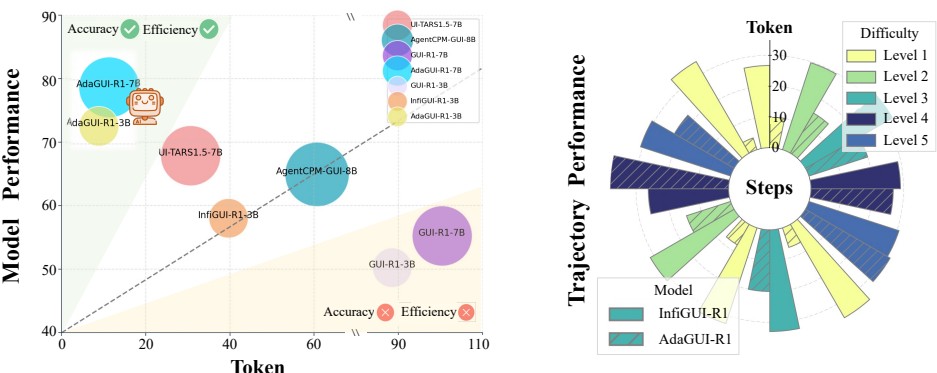

Figure 1: **AdaGUI-R1** embodies a human-like adaptive reasoning paradigm: it "thinks less" for easy steps and "thinks more" for hard ones, establishing a new state-of-the-art in efficiency (left). Within a specific trajectory with 12 steps, the number of reasoning token for each step is demonstrated starting from 12 o'clock clockwise, where step difficulty (the lower, the easier) is illustrated with different color (right). Our method reduces the number of reasoning tokens compared to models applying uniform-reasoning mechanism.

## 1 Introduction

By simulating the way users engage with graphical interfaces on phones, mobile GUI automation transforms manual processes into self-executing sequences, including steps like clicking buttons,

typing texts, scrolling screens, *etc*. With the help of mobile GUI automation, manual operations can be replaced and the efficiency of completing a task is improved. A few works have explored to achieve mobile GUI automation by constructing GUI Agents with elaborately annotated data. They typically map user instructions and pixels to actions such as *CLICK(x, y)* or *TYPE(text)*. However, their fundamental drawback lies in the absence of any explicit intermediate reasoning step, not articulating why the predicted action is appropriate. Consequently, these agents remain opaque "black-box": when they succeed, we cannot verify whether the choice was driven by genuine task understanding or spurious pixel correlations; when they fail, we receive no diagnostic trace beyond a mis-predicted action.

Inspired by DeepSeek-R1 (Guo et al., 2025; Shao et al., 2024), some researchers seek to guide the reasoning capability about the environment and actions in GUI agents using Group Relative Policy Optimization (GRPO). GUI-R1 (Luo et al., 2025) achieves contextual action prediction and verifiable reward-driven learning by designing a rule-based unified action space reward function among different platforms. InfiGUI-R1 (Liu et al., 2025) introduces sub-goal guidance and error recovery scenario construction to enhance reasoning for both planning and reflecting. Mobile-R1 (Gu et al., 2025) employs interactive multi-turn reinforcement learning with task-level rewards to enhance the exploration and error correction capabilities of GUI agents. MobileGUI-RL (Shi et al., 2025) proposes trajectory-aware policy optimization along with multi-component rewards to maintain a balance between task success rate and execution efficiency.

Despite dedicated efforts, these methods encourage reasoning equally in all steps without considering the significant differences of the step difficulty within a task. This uniform-reasoning strategy inevitably drags down not only efficiency, but also success rate in some cases (shown in Fig. 1). The performance degradation caused by uniform reasoning may be due to the following factors: 1) Some steps are trivial, such as clicking a clearly visible button or typing a piece of text. Yet they are forced through the same lengthy reasoning routine as steps that demand multi-conditional logic judgment, bringing about the risks of triggering over analysis and erroneously activating semantic associations of the model. 2) Some steps require only a straight-forward screenshot perception, whereas others need to integrating the task history to decide what to do next. Treating both cases identically wastes time on the easy ones and shortchanges the hard ones, dragging down both speed and accuracy.

To address the above limitations, we propose AdaGUI-R1, a GUI agent with difficulty-aware reasoning. Specifically, we first seed the agent with reasoning capability through supervised fine-tuning on a curated set of elaborate data, and further refine the reasoning capability with specially designed Group Adaptive Policy Optimization (GAPO) via difficulty-related rewards. To start with, each step in GUI automation tasks is evaluated by a pre-trained vision-language model to assess its difficulty. For steps that are marked as hard, we expand their original answers from bare *Action* to rich *Think-Action* pairs: the agent first emits an explicit Chain-of-Thought (CoT) that spells out why a specific action is necessary, then appends the concrete parseable *Action*. We employ a self-supervised CoT generation mechanism that yields reliable reasoning for actions and ensures the consistency between the reasoning and the final action to be executed. For steps that are marked as easy, their answers are simple prefixed "None" *Think* before the original *Action*. After the agent initially learns the reasoning paradigm, we enhance the difficulty-aware reasoning via reinforcement fine-tuning. During rollouts of each step, we calculate the difficulty by assessing the success rate within a group. Adaptive thought and action reward are applied for optimization based on the assessed difficulty. For steps tagged as high-difficulty, we grant a generous bonus whenever the agent produces a thoughtful CoT that ultimately leads to success, and an equally large penalty if it skips reasoning and fails. Conversely, for low-difficulty steps, we reward concise answers and lightly penalize answers with heavy deliberation, even when correct. Through the carefully designed pipeline, our GUI agent learns to think only when it truly matters, boosting both efficiency and task success. Furthermode, the adaptive action reward dynamically adjusts reward distributions based on action complexity, where simple actions require precise execution while complex actions benefit from broader exploration.

Our key contributions are summarized as follows:

- We introduce AdaGUI-R1, a mobile GUI agent that supports reasoning adaptively based on step difficulty. Benefiting from difficulty-aware reasoning, our method reduces the inference cost and increases the success rate of completing a GUI automation task.
- To ensure the quality of CoT during the initial injection of reasoning paradigm into our GUI agent, we devise a self-supervised CoT generation mechanism. This process is achieved by

using a pre-trained general multi-modal large language model without requiring any extra human labor, effectively ensuring the consistency between CoT and action to be executed.

- A novel Group Adaptive Policy Optimization algorithm for GUI automation tasks is proposed, which leverages difficulty-dependent thought reward as well as action exploration reward to encourage adaptive and accurate reasoning in reinforcement fine-tuning.

## 2 RELATED WORKS

**Mobile GUI Agent**  Mobile GUI Agents have evolved from API-dependent systems to vision-language model (VLM)-based agents capable of human-like interaction. Early systems like AppAgent (Zhang et al., 2025a; Jiang et al., 2025) focused on low-level rendering and limited functionality but struggled with generalization and unseen scenarios. Subsequent research integrated VLMs to perform GUI tasks, a series of improved GUI VLMs (Cheng et al., 2024; You et al., 2024; Li et al., 2024b; Papoudakis et al., 2025; Hong et al., 2024; Lin et al., 2024) are proposed to facilitate end-to-end action prediction with fundamental GUI perception. Meanwhile, multi-agent frameworks like Mobile-Agent (Wang et al., 2024) have introduced collaborative modules for perception, planning, memory, and execution, significantly improving generalization and accuracy in dynamic GUI environments (Rawles et al., 2024) .

Reinforcement learning (RL) has further augmented the reasoning capacities of GUI agents. DeepSeek-R1 (Guo et al., 2025; Shao et al., 2024) pioneered an RL framework aimed at strengthening reasoning in language models, inspiring a series of RL-enhanced GUI models including UI-R1 (Lu et al., 2025), GUI-R1 (Luo et al., 2025), and InfiGUI-R1 (Liu et al., 2025), each improving action prediction efficiency and task completion rates. GUI-Critic-R1 (Wanyan et al., 2025) extended this line by incorporating pre-operative error diagnosis, thus reducing execution mistakes prior to action commitment. A key differentiator among these approaches lies in their reward mechanisms: where some emphasize single-step accuracy, others, such as InfiGUI-R1, promote deliberative reasoning over reactive decision-making. Reward designs also vary considerably ranging from distance-based and IoU rewards in SE-GUI (Yuan et al., 2025) to more sophisticated grounding objectives in GUI-G$^2$ (Tang et al., 2025)—directly influencing agents' generalization across diverse and complex mobile environments.

**Adaptive Reasoning**  Growing computational demands and the prevalence of overthinking in reasoning models have motivated research into more efficient reasoning paradigms. A prominent line of work introduces adaptive reasoning mechanisms that modulate the depth of inference based on input complexity or model confidence. Zhang et al. (2025b) proposed AdaptThink, which uses proximal policy optimization (PPO) to supervise a policy that switches between no-thinking and thinking modes, effectively mitigating overthinking in language models. Similarly, AdaCoT (Lou et al., 2025; Huang et al., 2025) introduced an adaptive chain-of-thought approach that dynamically scales reasoning length according to perceived task difficulty. Follow-up work by Huang et al. (2025) extended this idea with reinforcement learning to trigger CoT only when beneficial. Further advancing this concept, ThinkSwitcher (Liang et al., 2025) trained a gating network to decide in real time whether to continue reasoning or halt, achieving improved efficiency without compromising performance. Waheed et al. (2025) score problem difficulty 1–10, compress CoT length to match, and distill with SFT+DPO. Yu et al. (2025) frame the same choice as a budget-allocation game, learning to lavish compute on hard queries and quit early on easy ones based on logits expectation. Wang et al. (2025) rank questions inside mini-batches to obtain a difficulty score and adaptively pick the number of sampled paths. Han et al. (2025) simply prompt the LLM to predict the minimal token budget before it starts. Aggarwal et al. (2023) early-stop majority voting once confidence (logits) is high, while Damani et al. trains a light helper that forecasts the marginal value of one more sample. Across methods, the recipe is identical: estimate difficulty, then spend just enough compute to get the answer. These methods are based on LLM and can not process visual input, thus they do not take the additional complexity brought by visual perception into account.

Our method focus on GUI automation and the definition of difficulty for adaptive reasoning differs from the above methods in mainly two aspects: 1) We determine the difficulty level based on the capabilities of the base model itself instead of relying on the inference of closed-source models or existing definitions in the dataset. 2) Under the circumstance of GUI instead of text only, a high logits only indicates a high level of confidence in the model's prediction, and does not necessarily

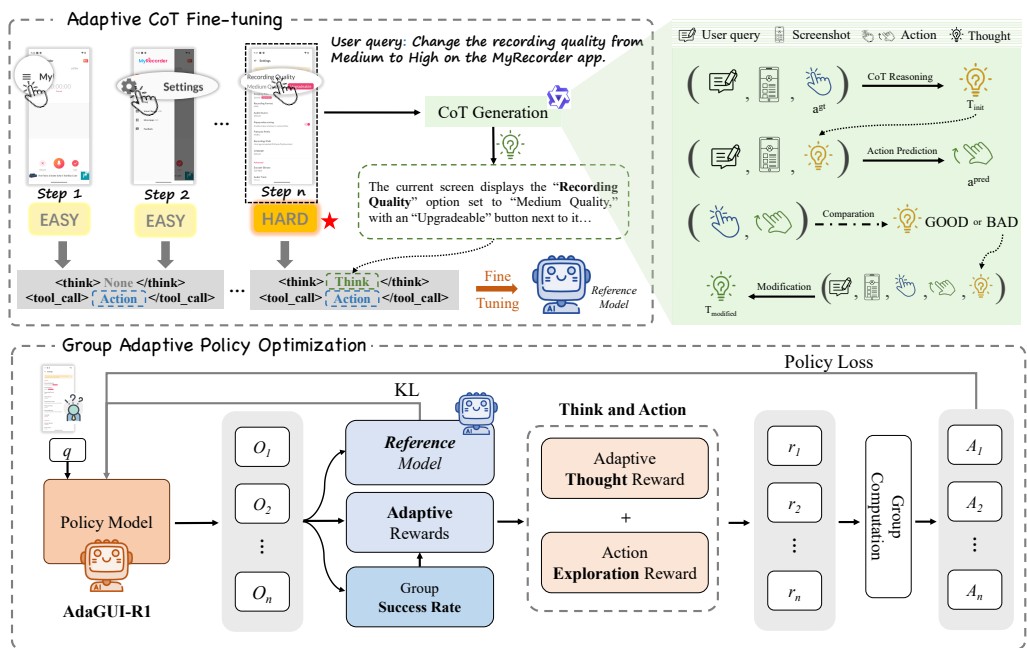

Figure 2: An overview of **AdaGUI-R1**. Adaptive CoT Fine-tuning initializes the agent with adaptive reasoning capabilities via supervised fine-tuning on a curated dataset including easy and hard steps. Subsequently, Group Adaptive Policy Optimization refines the agent's think and action ability via reinforcement fine-tuning using a adaptive thought reward as well as action exploration reward.

mean that the predicted action is correct. Our difficulty of a single step comes from the combined confidence level of image and text, which is more in line with real GUI interaction.

## 3 METHODS

In this section, we first introduce the effective GRPO algorithm in reinforcement fine-tuning as preliminary in Sec. 3.1, and then define the difficulty for steps as our foundation in Sec. 3.2. Later, we elaborate on how we induce the reasoning paradigms in Sec. 3.3, and further enhance reasoning capability via GAPO in Sec. 3.4. Our method is illustrated in Fig. 2.

### 3.1 PRELIMINARY

**Group Relative Policy Optimization (GRPO)** (Guo et al., 2025) is a reinforcement learning algorithm designed for training large language models, aiming to enhance model performance on complex tasks by optimizing the policy. The core idea of GRPO is to estimate the relative advantage of each response within a group of responses to the same query, thereby eliminating the need for a value function. The optimization objective can be expressed as:

$$J_{GRPO}(\theta) = \mathbb{E}_{q \sim P(Q), \{o_i\}_{i=1}^{G} \sim \pi_{\theta_{old}}(O|q)}$$

$$\frac{1}{G} \sum_{i=1}^{G} \left( \min \left( \frac{\pi_\theta(o_i|q)}{\pi_{\theta_{old}}(o_i|q)} A_{i,t}, \text{clip} \left( \frac{\pi_\theta(o_i|q)}{\pi_{\theta_{old}}(o_i|q)}, 1 - \epsilon, 1 + \epsilon \right) \right) - \beta \mathbb{D}_{KL}(\pi_\theta || \pi_{ref}) \right)$$

$$(1)$$

where $q$ denotes the query, $\mathbb{D}$ is the data distribution, $G$ is the number of responses generated for each query, $o_i$ is the $i$-th response, $\pi_{\theta_{old}}$ is the old policy, $\pi_\theta$ is the current policy, $A_{i,t}$ is the advantage of the $i$-th response at position $t$, and $\epsilon$ is the clipping range. $\mathbb{D}_{KL}(\pi_\theta || \pi_{\theta_{old}})$ denotes the KL penalty.

## 3.2 DIFFICULTY DEFINITION

Instead of relying on human annotators, we employ a pre-trained visual-language model to quantify the intrinsic difficulty of every step within a given task. Let the current GUI automation task be defined by a user query $Q$ and a set of screenshots $\{I_i\}$. At step $s$, the model is prompted with the concatenated context $P = [Q; I_s; \{a_0, a_1 ..., a_{s-1}\}]$, where $a_i$ denote previous action in step $i$. Given $P$, we perform $N$ independent forward passes with temperature-$\tau$ sampling, yielding the action set ($N$ is set to 10 empirically):

$$\mathbb{A}_s = \{a_s^j\}_{j=0}^N, \quad a_s^j \sim \text{model}(\cdot | P, \tau). \tag{2}$$

By comparing these predicted actions against the ground-truth action, we can compute a simple accuracy score $\rho_s$ for step $s$:

$$\rho_s = \frac{1}{N} \sum_{j=0}^{N} \mathbb{1}[a_s^j = a_s^{gt}], \tag{3}$$

where $a_s^{gt}$ is the ground-truth action. A discrete difficulty level $\ell \in \{1, \ldots, 5\}$ is assigned via:

$$\ell(\rho) = 5 - \max(0, \lceil 5\rho \rceil - 1) \quad \text{with} \quad \rho \in [0, 1]. \tag{4}$$

Empirically, steps with a difficulty level less than or equal to 3 are classified as "easy", while those with a level greater than 3 are categorized as "hard". In this way, model's own uncertainty is manifested through prediction disagreement, acting as a principled proxy for step difficulty.

## 3.3 INDUCING REASONING PARADIGM VIA ADAPTIVE CoT FINE-TUNING

Since the difficulty in GUI automation tasks varies among steps, we encourage the model to think more in hard steps and less in easy steps when reasoning. To seed the initial difficulty-aware reasoning capability into our GUI agent, we must first teach it "how" and "when" to generate CoT. We therefore curate some annotated data in which every hard step is paired with full CoT why an action should be executed, and simple steps are paired with empty CoT for format alignment. (Note that "model" in Eq. 2 is our base model in this stage.) Next, we introduce our method of obtaining full CoT via self-supervised CoT generation.

Rather than naively distilling the first-pass CoT generated by a multi-modal large language model, we introduce a self-supervised pipeline for CoT generation, thereby yielding higher-quality supervisory data for inducing reasoning paradigm of our GUI agent. We first feed the user query $Q$ together with the current screenshot $I$ into a multi-modal large language model and provide the ground-truth next action $a^{gt}$ as an explicit hint. Conditioned on this hint, the model is then asked to generate the first-pass CoT $T_{init}$ that would rationally lead to conducting that action. Subsequently, the first-pass CoT $T_{init}$ is injected as an in-context prompt, enabling the multi-modal large language model to predict the next action $a^{pred}$ conditioned on both the user query $Q$ and the current screenshot $I$. Next, the predicted action $a^{pred}$ is then compared with the ground-truth one $a^{gt}$. The multi-modal large language model is prompted to assess their equivalence; if they match, the first-pass CoT $T_{init}$ is retained, otherwise the model re-analyses the user instruction, the current screenshot, and the mismatch, and outputs a revised CoT that replaces the previous one as output. We show our intuitive implementation in Algorithm 1.

With this self-supervised CoT generation mechanism, we obtain the content of *Think* for hard steps. And we simply left *Think* blank for easy steps. Thus, the answer of training data can be expand from bare *Action* to *Think-Action* pairs. The *Think-Action* pairs are then distilled into our GUI agent via supervised fine-tuning, through which our GUI agent learns explicit reasoning. In this way, the trained agent learns to generate CoT when the current step is hard and directly predict action when the current step is easy. Our initially trained agent then becomes the foundation on which later difficulty-aware reinforcement refinements are built.

## 3.4 ENHANCING REASONING VIA GROUP ADAPTIVE POLICY OPTIMIZATION

In mobile agent reinforcement learning, algorithms like GRPO (Guo et al., 2025) are commonly used to optimize policies based on step-level rewards. However, conventional reward mechanisms

---

**Algorithm 1** Self-Supervised CoT Generation

---

**Input:** User query $Q$; current screenshot $I$; ground-truth action $a^{gt}$.
**Output:** CoT $T_{final}$.
**Model:** Multi-modal large language model (MLLM).
1: $T_{init} \sim \text{MLLM}(Q, I, a^{gt})$         ▷ Generate first-pass CoT based on ground-truth action
2: $a^{pred} \sim \text{MLLM}(Q, I, T_{init})$         ▷ Generate prediced action based on first-pass CoT
3: $judgement \sim \text{MLLM}(a^{gt}, a^{pred})$         ▷ Obtain equivalence as true/false
4: **if** $judgement$ is true **then**
5:     $T_{final} \leftarrow T_{init}$         ▷ Keep first-pass CoT
6: **else**
7:     $T_{final} \sim \text{MLLM}(Q, I, a^{gt}, a^{pred}, T_{init})$         ▷ Reanalyse and modify CoT
8: **end if**

---

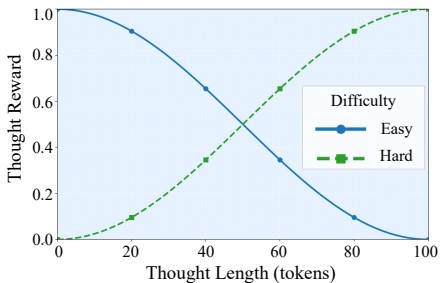

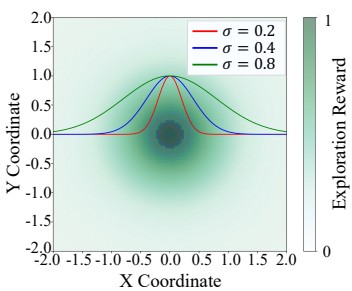

Figure 3: Visualization of thought reward. The reward value is modeled as a cosine function of thought length, with opposing gradients for easy and hard tasks.

Figure 4: Gaussian exploration reward for click actions. The spatial distribution of the exploration reward is modulated by the step's difficulty, explicitly represented by $\sigma$ values.

face two significant challenges. First, **redundant thinking processes** often lead to decreased reasoning accuracy and efficiency. Second, the binary nature of the action reward (*e.g.*, 1.0 for success, 0 for failure) creates a **lack of reward variance**. When all successful actions within a sampled group receive an identical reward of 1.0, the model cannot distinguish between a precise, confident action and a marginally successful one. This uniform reward signal is particularly problematic for complex operations where nuanced feedback is essential for effective exploration. To address these limitations, we propose the *Group Adaptive Policy Optimization (GAPO)* algorithm, which introduces a more sophisticated and adaptive reward structure including thought and exploration reward. (Note that "model" in Eq. 2 is our iteratively update model in this stage.)

### 3.4.1 ADAPTIVE THOUGHT REWARD

In practical use, human *do not* think or perform equally for every step:

- **Intuitive actions** (e.g., tapping "Back", closing the keyboard) are executed almost instantly with little conscious thought.

- **Complex actions** (e.g., calibrating a colour-picker, filling a multi-field form) are preceded by visible hesitation, planning, or mental simulation.

Currently, existing reward-shaping approaches for "think" only impose simple length constraints on reasoning, lacking alignment with human cognitive patterns and failing to encourage difficulty-aware reasoning strategies. To bridge this gap, we propose an *Adaptive Thought Reward* mechanism that mimics human's intuition as a differentiable training signal. Its core principle is: *Penalize short thinking for easy operations, encourage long thinking for hard ones.*

The adaptive thought reward is computed based on the length $t$ of the thought sequence and the difficulty level $\ell$:

$$R_{\text{thought}}(t \mid \ell) = \begin{cases} \max -(\max - \min) \cdot \dfrac{1 - \cos(\pi t/T)}{2}, & \ell \leq \ell_{thr} \quad \text{(easy)} \\ \min +(\max - \min) \cdot \dfrac{1 - \cos(\pi t/T)}{2}, & \ell > \ell_{thr} \quad \text{(hard)} \end{cases} \tag{5}$$

where $T$ is a maximum thought length, and $\min = 0.01$, $\max = 1.0$ define the bounded reward range. As visualized in Fig. 3, this function is smooth, differentiable, and strictly monotonic with respect to $t$ within the interval $[0, T]$. This design encourages the emergence of adaptive thinking patterns: the policy learns to produce minimal thought for trivial or mastered operations ($\ell \leq \ell_{thr}$), while engaging in extended reasoning for challenging and error-prone steps ($\ell > \ell_{thr}$), effectively aligning the model's cognitive effort with the underlying task difficulty. ($\ell_{thr}$ is set to 3 empirically.)

### 3.4.2 Action Exploration Reward

In GUI task automation, traditional reinforcement learning approaches often suffer from *sparse reward signals* , where the agent only receives positive feedback when executing perfectly correct actions. This sparse reward structure poses significant challenges for learning complex operations. To address this issue, we introduce an action exploration reward, particularly for prevalent actions *Click*. Instead of a uniform reward, this reward is spatially modeled as a difficulty-adaptive Gaussian distribution centered on the ground-truth coordinates. For easy steps, a narrow Gaussian enforces high precision. For hard steps, a wider Gaussian provides partial credit for "near misses", creating a richer reward landscape that guides the agent to explore more effectively in challenging scenarios.

**Difficulty-aware bandwidth** We let the kernel width *self-adapt* to the instantaneous difficulty of the click step. With the per-step difficulty level $\ell$, the adaptive standard deviation is set as

$$\sigma(\ell) = \sigma_{\min} + \Delta\sigma \cdot \frac{\ell}{5}, \quad \Delta\sigma = \sigma_{\max} - \sigma_{\min}, \tag{6}$$

where $\sigma_{\min} = 0.2$ and $\sigma_{\max} = 0.8$ are normalized with respect to the size of the centered element.

**Gaussian exploration reward** Let $\boldsymbol{\mu}_{\text{gt}} = (x_{\text{gt}}, y_{\text{gt}})$ be the ground-truth center and $\boldsymbol{p} = (x_{\text{pred}}, y_{\text{pred}})$ the predicted click point. The difficulty-conditioned Gaussian reward function $R_{\text{point}}(\boldsymbol{p} \mid \ell)$ is defined as follows:

$$R_{\text{point}}(\boldsymbol{p} \mid \ell) = \begin{cases} \exp\left(-\frac{\|\boldsymbol{p} - \boldsymbol{\mu}_{\text{gt}}\|^2}{2\sigma^2(\ell)}\right), & \text{if } \|\boldsymbol{p} - \boldsymbol{\mu}_{\text{gt}}\| \leq N_{thr}; \\ 0, & \text{otherwise}; \end{cases} \tag{7}$$

where the Euclidean distance between the predicted and ground truth positions is calculated as:

$$\|\boldsymbol{p} - \boldsymbol{\mu}_{\text{gt}}\|^2 = \frac{(x_{\text{pred}} - x_{\text{gt}})^2}{w^2} + \frac{(y_{\text{pred}} - y_{\text{gt}})^2}{h^2}. \tag{8}$$

The $N_{\text{thr}}$ signifies the threshold value representing the relative width of the click operation tolerance, set to $0.04$ empirically. $w$ and $h$ denote the actual width and height of the screen in pixels, respectively. $\sigma(\ell)$ represents the standard deviation (or width of the Gaussian) for the Gaussian function, which varies dynamically with the difficulty level $\ell$. Gaussian exploration rewards with various $\sigma(\ell)$ parameters are visually displayed in Fig. 4. For easy steps, they have narrow kernels ($\sigma_{min}$) and sharp peaks, encouraging high precision. For hard steps, they have wide kernels ($\sigma_{max}$) and smooth plateau, providing dense gradients for difficult click operation.

## 4 Experiments

### 4.1 Experiments Setting

**Datasets and Benchmark** To reduce computational cost while maintaining benchmark consistency, we use half randomly sampled data from the official training sets of Amex (Chai et al., 2024), AITZ (Zhang et al., 2024), Android Control (Li et al., 2024a), and GUI Odyssey (Lu et al., 2024). The total size of our training data is 116.5k. The evaluation is carried out on the official test sets

Table 1: Experimental results for different models across various datasets. TM and EM refers to average action type match rate and average action parameter match rate. Tok. represents the average thinking tokens generated by reasoning models. "-" denotes invalid results. Base-3B and Base-7B denotes Qwen2.5-VL-3B and Qwen2.5-VL-7B, respectively.

| Model | GUI Odyssey | | | AITZ | | | Android Control | | |
|---|---|---|---|---|---|---|---|---|---|
| | TM | EM | Tok. | TM | EM | Tok. | TM | EM | Tok. |
| *Commercial Models* | | | | | | | | | |
| GPT-4o(Achiam et al., 2023) | - | 20.39 | - | 70.00 | 35.30 | - | - | 20.80 | - |
| Claude (Anthropic, 2024) | 60.90 | - | - | - | - | - | - | 12.50 | - |
| Gemini(Team et al., 2024) | - | 3.27 | - | - | - | - | - | 60.20 | - |
| *General Open-source Models* | | | | | | | | | |
| Qwen2.5-VL-3B(Bai et al., 2025) | 60.23 | 44.65 | - | 75.14 | 52.73 | - | 76.53 | 60.17 | - |
| Qwen2.5-VL-7B(Bai et al., 2025) | 59.54 | 46.28 | - | 78.41 | 54.61 | - | 75.10 | 62.90 | - |
| *GUI-specific Models (w/o reasoning)* | | | | | | | | | |
| Aguvis-7B(Xu et al., 2024) | 26.71 | 13.54 | - | 35.71 | 18.99 | - | 65.56 | 54.18 | - |
| OS-Genesis-7B(Sun et al., 2024) | 11.67 | 3.63 | - | 19.98 | 8.45 | - | 65.92 | 44.43 | - |
| OS-Atlas-Pro-7B(Wu et al., 2024) | 91.83 | 76.76 | - | 74.13 | 58.45 | - | 70.36 | 56.53 | - |
| OdysseyAgent-7B(Lu et al., 2024) | 90.83 | 73.67 | - | 59.17 | 31.60 | - | 58.80 | 32.74 | - |
| *GUI-specific Models (w reasoning)* | | | | | | | | | |
| GUI-R1-3B(Luo et al., 2025) | 56.29 | 50.10 | 88.54 | 50.55 | 44.35 | 92.63 | 52.14 | 47.95 | 95.58 |
| InfiGUI-R1-3B(Liu et al., 2025) | 74.50 | 55.02 | 39.60 | 70.77 | 52.88 | 43.99 | 71.53 | 58.04 | 42.30 |
| GUI-R1-7B(Luo et al., 2025) | 63.21 | 55.26 | 113.37 | 56.75 | 50.47 | 118.02 | 57.48 | 51.88 | 116.06 |
| UI-TARS-7B(Qin et al., 2025) | 86.06 | 67.90 | 23.66 | **80.42** | 65.77 | 22.57 | 75.10 | 62.90 | 22.88 |
| **AdaGUI-R1-3B** | **87.10** | **72.97** | **7.85** | 75.89 | 59.91 | 8.51 | 81.91 | 69.74 | 8.16 |
| Δ (vs Base-3B) | *+26.87* | *+28.32* | *-* | *+0.75* | *+7.18* | *-* | *+5.38* | *+9.57* | *-* |
| **AdaGUI-R1-7B** | **89.24** | **77.88** | **11.26** | 78.64 | **66.62** | 14.67 | 82.97 | 72.40 | 13.19 |
| Δ (vs Base-7B) | *+29.70* | *+31.60* | *-* | *+0.23* | *+12.01* | *-* | *+7.87* | *+9.50* | *-* |

without any modification on three mobile GUI automation benchmarks, including AITZ, Android Control, and GUI Odyssey. Each benchmark adopts three evaluation metrics: **Type Match (TM)**, which verifies that the predicted action type agrees with the ground truth, **Exact Match (EM)**, which further demands that every parameter be predicted without error, and **Tok.**, which counts the number of average thinking tokens in reasoning models.

**Parameter Settings** We adopt Qwen2.5-VL-3B and Qwen2.5-VL-7B as our base model of different scales. LoRA (Hu et al., 2022) is applied to fine-tuned the base models, with lora-rank $r$ and lora-alpha $\alpha$ set to 32 and 64 in adaptive CoT fine-tuning, 8 and 32 in GAPO reinforcement fine-tuning. The learning rate for two fine-tuning stages are set to $5e^{-5}$ and $1e^{-6}$ respectively. All experiments are conducted on $8\times$NVIDIA A100-80G GPUs.

## 4.2 MAIN RESULTS

The performance of our models against state-of-the-art baselines is detailed in Tab. 1. As presented in Tab. 1, our AdaGUI-R1 framework yields consistent and substantial gains over the base models on all three benchmarks. The results underscore the value of difficulty-aware reasoning for robust GUI automation. Our method improves the success rate of GUI automation tasks for 3B and 7B models, with the average success rate increasing from 52.51 to 67.54 at 3B scale. Compared to GUI-R1 (Luo et al., 2025), which is concurrent work also trained using reinforcement fine-tuning, our model achieves an over 20-point improvement at 3B scale and an around 20-point improvement at 7B scale, validating that concentrating reasoning effort on challenging steps not only improves the accuracy of hard steps, but also avoids the hallucination triggered by introducing over analysis in easy steps. As for efficiency, AdaGUI-R1, a difficulty-aware reasoning agent, reduces the average thinking token numbers by 40% compared to other uniformly reasoning models, which can result in faster inference and shorter user-perceived response times. Overall, our method achieves both efficient and effective GUI automation by our design of thinking only when necessary.

## 4.3 ABLATION STUDY

We conduct a series of ablation studies to systematically evaluate the contribution of each key component within our AdaGUI-R1 framework. Specifically, we investigate: 1) the effectiveness of

Table 2: Ablation study of different CoTs and rewards on Android Control benchmark. Using self-supervised and difficulty-aware CoT along with full rewards shows the best performance.

| Model | Android Control | | | |
|---|---|---|---|---|
| | **EM** | $\Delta_{\textbf{EM}}$ | **Tok.** | **Ratio$_{\textbf{Tok.}}$** |
| **AdaGUI-R1-7B (Full)** | 72.40 | - | 13.19 | - |
| *CoT Ablation* | | | | |
| -w/o self-supervised CoT | 65.50 | **-6.82%** | 13.04 | **0.99$\times$** |
| -w/o difficulty-aware CoT | 62.84 | **-9.48%** | 34.18 | **2.59$\times$** |
| *Rewards Ablation* | | | | |
| -w/o Thought Reward | 69.52 | **-2.80%** | 29.81 | **2.26$\times$** |
| -w/o Exploration Reward | 68.47 | **-3.85%** | 13.92 | **1.06$\times$** |
| -w/o Exploration and Thought Reward | 65.12 | **-7.2%** | 28.43 | **2.16$\times$** |

Table 3: Ablation study of different scales of difficulty definition on Android Control benchmark. Five-tier scale facilitates the action exploration to be more precise and stable, showing the best performance.

| Model: AdaGUI-R1-7B | Android Control | | | |
|---|---|---|---|---|
| | **EM** | $\Delta_{\textbf{EM}}$ | **Click ACC** | $\Delta_{\textbf{Click-ACC}}$ |
| **5-tier scale** | 72.40 | - | 75.16 | - |
| 2-tier scale | 69.20 | **-4.42%** | 71.84 | **-4.42%** |
| 10-tier scale | 70.16 | **-3.09%** | 73.70 | **-1.94%** |

self-supervised CoT, 2) the way of inducing reasoning paradigm, 3) the effectiveness of GAPO algorithm, and 4) the parameters in rewards. We benchmark all variants on Android Control. The results are summarized in Tab. 2 and presented in Figure 5.

**Effectiveness of self-supervised CoT**  To validate the effectiveness of our self-supervised CoT generation mechanism, we conduct an experiment in which first-pass CoT $T_{init}$ in Algorithm 1 is used for adaptive CoT fine-tuning. As shown in the second row of Tab. 2, directly using $T_{init}$ as reasoning can lead to poor performance, as there may be inconsistencies between the thinking process and the actions to be executed. By using self-supervised CoT generation mechanism to further analyze and correct the cot, the quality of the CoT injected into the model can be improved, laying a good foundation for the subsequent improvement of reasoning ability.

**The way of inducing reasoning paradigm**  To extend the original output of the agent from bare *Action* to *Think-Action* pairs, the reasoning process should be added. We conduct a ablation study that applied self-supervised CoT to all steps rather than only to defined hard steps. The third row of Tab. 2 indicates that this "all-CoT" policy during supervised fine-tuning stage can lead to a deterioration in the perception of step difficulty, which results in suboptimal performance for subsequent reasoning enhancement. Our "difficulty-aware" CoT policy validates that adaptive reasoning is both sufficient and superior.

**Effectiveness of Adaptive Rewards**  Our analysis examines the impact of Thought Reward and Exploration Reward on model performance. As shown in the forth tow of Tab. 2, removing Exploration Reward results in a slight decrease in accuracy and a minimal increase in token usage, indicating their contribution to accuracy through effective exploration. Conversely, the absence of Thought Reward (the fifth row in Tab. 2) causes a minor accuracy drop but a significant token increase, highlighting their role in optimizing model efficiency by promoting strategic thinking and reducing unnecessary actions.

**Level Definition Ablation**  In our main experiments, we define the discrete difficulty level in a five-tier scale in Eq. 4. This difficulty level mainly effects the Gaussian bandwidth in action exploration. We carried out ablation studies on different scales of level, *i.e.* 2 and 10. For two-tier scale, the

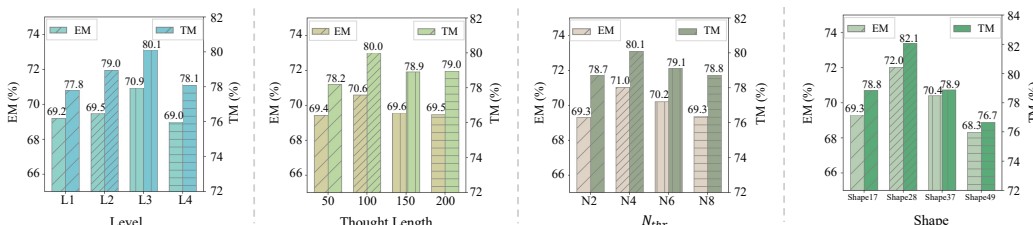

Figure 5: Ablation studies on key hyperparameters of our model. We evaluate the impact when varying (from left to right): difficulty level definition, maximum thought length, gaussian normalization width, and gaussian reward shape.

mapping funtion is $\ell(\rho) = 2 - \lceil \rho - 0.5 \rceil$. For ten-tier scale, the mapping funtion is $\ell(\rho) = 11 - \lceil 5\rho + 0.5 \rceil$. From Tab. 3, five-tier scale of difficulty level outperforms others, indicating that this setting facilitates the action exploration to be more precise and stable.

**Rewards Parameter Ablation** We performed an ablation study to assess the impact of specific Reward parameter settings on performance, with results visualized in Fig. 5. For Thought Reward, we considered the maximum thought length and switching difficulty level, which dictate the thinking mode. Experiments indicated that switching modes at *Level* = 3 and maintaining a maximum thought length of 100 tokens optimizes model inference accuracy. For Exploration Reward, we evaluated the Gaussian function's cutoff point and shape across difficulty levels, where Shape$AB$ refers to gaussian shape with $\sigma_{min} = A/10$ and $\sigma_{max} = B/10$. N$A$ refers to gaussian normalized threshold as $N_{thr} = A/100$. Our setting as $\sigma_{min} = 0.2$, $\sigma_{max} = 0.8$, and $N_{thr} = 0.04$ yields the highest prediction accuracy, confirming the efficacy of our multi-level Gaussian Reward approach.

# 5 LIMITATION

Our method is designed on offline scenarios which concentrate on step-level accuracy. Online scenario is inherently more challenging: user behaviors are noisy, task boundaries are blurred, no ground truth actions could be used for history, the reflective ability of agents needs to be strengthened. A reflective mechanism that can pause, revise, or rollback its previous decisions—without access to ground-truth actions is still missing in the present method.

We therefore regard bridging the offline-online gap as the principal challenge of our future works: equipping the agent with self-corrective reflection that operates under noisy, label-free, streaming conditions.

# 6 CONCLUSION

In this work, we proposed *AdaGUI-R1*, a novel GUI automation agent that employs difficulty-aware reasoning to enhance both task execution efficiency and accuracy. The integration of a self-supervised CoT generation mechanism ensures high-quality reasoning trajectories, while a Group Adaptive Policy Optimization algorithm further refines adaptive reasoning. Results across diverse benchmarks and various scales demonstrate that AdaGUI-R1 not only achieves superior task success but also reduces average thinking tokens, delivering markedly faster inference. Looking ahead, the principles underlying AdaGUI-R1 could be extended to real-time GUI tasks requiring instant decision-making processes, paving the way for more intelligent automation solutions.

STATEMENT OF USING LLM

We use LLM to help us with converting our equations and tables into LATEX format.

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

# A APPENDIX

## A.1 ACTION DISTRIBUTION

Tab. 4 lists the top-3 action types and their ratios across datasets; *Click* dominate every split, consistently exceeding 50% of all steps. This heavy-tailed distribution confirms that *Click* is the de-facto atomic action in mobile GUI automation, which motivates us to apply deeper exploration on *Click* to improve the performance of our GUI agent.

Table 4: Top-3 Actions and corresponding distribution for our train datasets.

| Dataset | GUI Odyssey | | | Amex | | |
|---|---|---|---|---|---|---|
| **Top-3 Actions** | Click | Type | Swipe | Click | Swipe | Terminate |
| **Ratio(%)** | 65.34 | 10.59 | 9.22 | 64.38 | 19.45 | 7.61 |
| **Dataset** | AITZ | | | Android Control | | |
| **Top-3 Actions** | Click | Terminate | Type | Click | Terminate | Swipe |
| **Ratio(%)** | 53.94 | 17.46 | 11.66 | 52.63 | 15.45 | 11.06 |

## A.2 DIFFICULTY LEVEL DISTRIBUTION

Fig. 6 provides a comparative analysis of level distributions across four datasets(AITZ, Android Control, Odyssey, and Amex).

Fig. 7 illustrates the level change before and after the implementation of Reinforce-Fine-Tuning (RFT). The application of RFT results in an increase in Level 1 and Level 2 data and a decrease in Level 5 tasks, suggesting a shift towards handling more basic steps efficiently.It apparently refines the model's ability to tackle tasks of varying difficulty levels, with a general trend towards better handling of both simple and complex tasks. This indicates that RFT effectively enhances the model's adaptability to step difficulty.

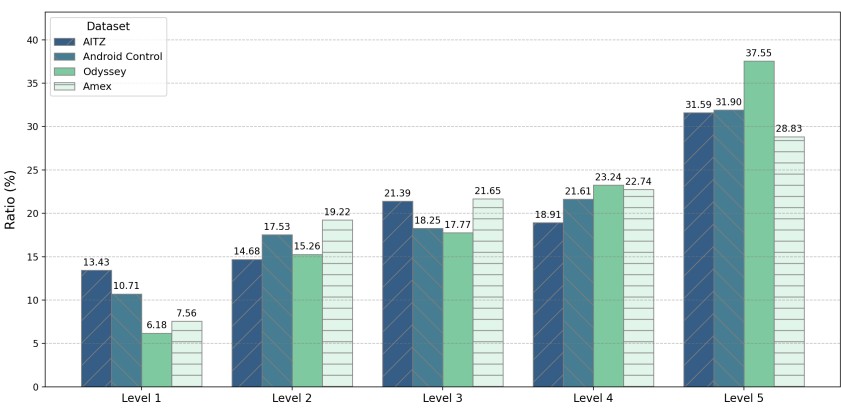

Figure 6: Distribution of difficulty levels across different datasets.

## A.3 ACTION SPACE AND RESPONSE FORMAT

We adopt a unified action space to ensure that all task-level instructions can be decomposed into a sequence of atomic actions via <tool call>. There are eight actions: key, click, swipe, long press, type, system button, terminate, and wait. (Rawles et al., 2023).Following the pioneers, our format response as <think>,<tool call>.

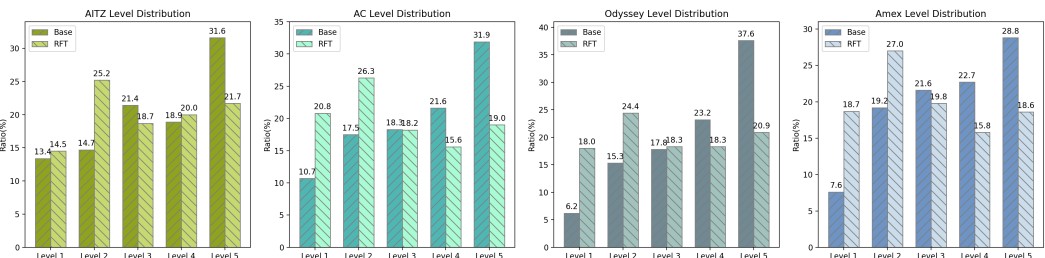

Figure 7: Distribution of levels for different datasets before and after RFT

## A.4 PROMPT FOR TRAINING AND INFERENCE

**System prompt:**
You FIRST decide whether the current screenshot requires a thinking process to achieve the user query.
If so, first think about the reasoning process as an internal monologue and then provide the final answer. The reasoning process MUST BE enclosed within `<think>` `</think>` tags.
If not, no reasoning process needs to be enclosed within `<think>` `</think>` tags, and provide the final answer.
*Tools description (detailed in A.5).*
For each function call, return a json object with function name and arguments within `<tool_call></tools_call>` XML tags:
`<tool_call>`
{"name": <function-name>, "arguments": <args-json-object>}
`</tools_call>`

**User prompt:**
The user query: USER INSTRUCTION
Task progress (You have done the following operation on the current device): HISTORY ACTIONS

## A.5 Tools Description in Prompt

**Tools**

You may call one or more functions to assist with the user query.

You are provided with function signatures within `<tools></tools>` XML tags:

```
<tools>{
  "type": "function",
  "function": {
    "name": "mobile_use",
    "description":
      "Use a touchscreen to interact with a mobile device, and take screenshots.
      * This is an interface to a mobile device with touchscreen. You can perform actions like
      clicking, typing, swiping, etc.
      * Some applications may take time to start or process actions, so you may need to wait
      and take successive screenshots to see the results of your actions.
      * The screen's resolution is WIDTH×HEIGHT.
      * Make sure to click any buttons, links, icons, etc with the cursor tip in the center of the
      element. Don't click boxes on their edges unless asked.",
    "parameters": {"properties":
      {"action": {
        "description": "The action to perform. The available actions are:
          * key: Perform a key event on the mobile device.
            - This supports adb's 'keyevent' syntax.
            - Examples: "volume_up", "volume_down", "power", "camera", "clear".
          * click: Click the point on the screen with coordinate (x, y).
          * long_press: Press the point on the screen with coordinate (x, y) for specified
          seconds.
          * swipe: Swipe from the starting point with coordinate (x, y) to the end point with
          coordinates2 (x2, y2).
          * type: Input the specified text into the activated input box.
          * system_button: Press the system button.
          * open: Open an app on the device.
          * wait: Wait specified seconds for the change to happen.
          * terminate: Terminate the current task and report its completion status.",
        "enum": [key, click, long_press, swipe, type, system_button, open,
        wait, terminate], "type": "string"},
      "coordinate": {"description": "(x, y): The x (pixels from the left edge) and y (pixels
      from the top edge) coordinates to move the mouse to. Required only by action=click,
      action=long_press, and action=swipe.", "type": "array"},
      "coordinate2": {"description": "(x, y): The x (pixels from the left edge) and y (pixels
      from the top edge) coordinates to move the mouse to. Required only by action=swipe.",
      "type": "array"},
      "text":    {"description":    "Required   only   by   action=key,   action=type,   and
      action=open.", "type": "string"},
      "time": {"description": "The seconds to wait. Required only by action=long_press
      and action=wait.", "type": "number"},
      "button": {"description": "Back means returning to the previous interface, Home means
      returning to the desktop, Menu means opening the application background menu, and
      Enter means pressing the enter. Required only by action=system_button", "enum":
      [Back, Home, Menu, Enter], "type": "string"},
      "status": {"description": "The status of the task. Required only by action=terminate.",
      "type": "string", "enum": [success, failure]}},
    "required": ["action"], "type": "object"}}}
</tools>
```

## A.6 T-SNE VISUALIZATION OF TRAINING AND TEST SETS

We map samples (instructions+screenshots) from the official training set and test sets to the 2D t-SNE space using Qwen2.5-VL-7B feature extractor, as shown in Fig. 8. The similarity in distribution between training and testing points in 2D space rather than clustering separation is mainly due to the natural concentration of task types. After verification, there is no cross set duplication in the screenshot file name and instruction text in official independent partitions.

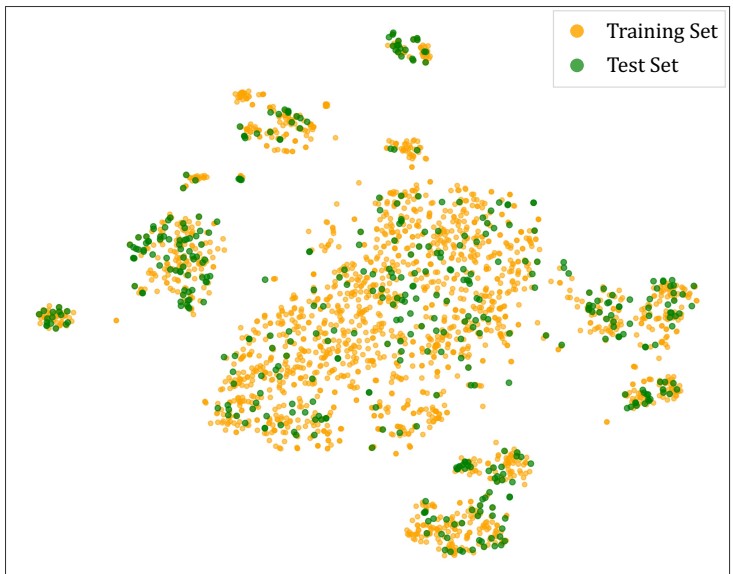

Figure 8: Distribution of official training test sets in a unified t-SNE space.

## A.7 MORE QUANTITATIVE RESULTS

We additionally report the confidence interval of GUI-specific reasoning models in Tab. 5 as a supplement of Tab. 1. The accuracy and its 95% confidence interval computed with the Wilson score interval over each test set is presented.

Table 5: Statistical significance of differences between AdaGUI-R1 and baselines.

| Model | GUI Odyssey | | AITZ | | Android Control | |
|---|---|---|---|---|---|---|
| | TM | EM | TM | EM | TM | EM |
| GUI-R1-3B | 56.29($\pm$0.57) | 50.10($\pm$0.57 ) | 50.55($\pm$1.43) | 44.35($\pm$1.42) | 52.14($\pm$0.97) | 47.95($\pm$0.97) |
| InfiGUI-R1-3B | 74.50($\pm$0.50) | 55.02($\pm$0.57) | 70.77($\pm$1.30) | 52.88($\pm$1.42) | 71.53($\pm$0.88) | 58.04($\pm$0.96) |
| GUI-R1-7B | 63.21($\pm$0.55) | 55.26($\pm$0.57) | 56.75($\pm$1.41) | 50.47($\pm$1.43) | 57.48($\pm$0.96) | 51.88($\pm$0.97) |
| UI-TARS-7B | 86.06($\pm$0.40) | 67.90($\pm$0.53) | 80.42($\pm$1.13) | 65.77($\pm$1.35) | 75.10($\pm$0.84) | 62.90($\pm$0.94) |
| AdaGUI-R1-3B | 87.10($\pm$0.38) | 72.97($\pm$0.50) | 75.89($\pm$1.22) | 59.91($\pm$1.40) | 81.91($\pm$0.75) | 69.75($\pm$0.89) |
| AdaGUI-R1-7B | 89.24($\pm$0.35) | 77.88($\pm$0.47) | 78.64($\pm$1.17) | 66.62($\pm$1.34) | 82.97($\pm$0.73) | 72.40($\pm$0.87) |

## A.8 MORE ABLATION RESULTS

We have considered different designs of our adaptive reward during enhancing reasoning.

For adaptive thought reward, instead of using cosine-like function as in Eq. 5, we apply easier deigns like delta-like function (Eq. 9) or continuous function based on step accuracy score (Eq. 10):

$$R_{\text{thought-delta}}(t \mid \ell) = \begin{cases} \mathbb{1}[t < T/2], & \ell \leq \ell_{thr} \quad \text{(easy)} \\ \mathbb{1}[t > T/2], & \ell > \ell_{thr} \quad \text{(hard)} \end{cases} \tag{9}$$

$$R_{\text{thought-acc-based}}(t, \rho) = (1 - \rho) * (1 - e^{-t/T}) + \rho * e^{-t/T}, \tag{10}$$

where $T$ is a maximum thought length, $\ell_{thr}$ is set to 3, and $\rho$ is defined in Eq. 3. As shown in the second row in Tab. 6, delta-like thought reward brings in severe reward variation at the threshold, which would cause instability of training, resulting in superior performance. Meanwhile, continuous accuracy-based thought reward causes fuzzy signals for mid-difficulty steps ($\rho \in [0.4, 0.6]$), resulting in increase of reasoning token.

For action exploration reward, we replace the designed Gaussian reward (Eq. 7) with box-like action reward:

$$R_{\text{point}}(p) = \begin{cases} 1, & \text{if } p \text{ in element box,} \\ 0, & \text{else.} \end{cases} \tag{11}$$

The forth row of Tab. 6 demonstrates that the box-like action reward lacks varying degrees of feedback on the different positions of points falling within the box, a precise control of click actions has decreased, resulting in reduced performance.

Table 6: Ablation study of different designs on thought and action rewards on Android Control benchmark. Using cosine-like thought reward and Gaussian action reward shows the best performance.

| Model | Android Control | | | |
| --- | --- | --- | --- | --- |
| | EM | $\Delta_{\text{EM}}$ | Tok. | Ratio$_{\text{Tok.}}$ |
| **AdaGUI-R1-7B (cosine-like thought reward & Gaussian action reward)** | 72.40 | - | 13.19 | - |
| *Thought Reward Design Ablation* | | | | |
| -delta-like thought reward | 70.27 | **-2.94%** | 15.12 | **1.15×** |
| -accuracy-based thought reward | 72.03 | **-0.51%** | 16.92 | **1.28×** |
| *Action Reward Design Ablation* | | | | |
| -box-like action reward | 69.93 | **-3.41%** | 13.27 | **1.01×** |

A full task success rate metric refers to the proportion of trajectories in which each step is successfully predicted. We additionally provide the full task success rate of our full and ablation models in Tab. 7, which proves the effectiveness of our method in another aspect.

Table 7: Full task success rate of full model and ablation ones, which calculated the ratio of trajectories where the prediction of all steps are correct.

| Model | GUI Odyssey | AITZ | Android Control |
| --- | --- | --- | --- |
| **AdaGUI-R1-7B (full)** | **15.62** | **7.91** | **25.12** |
| -w/o self-supervised CoT | 10.91 | 3.01 | 16.27 |
| -w/o difficulty-aware CoT | 9.42 | 3.26 | 14.61 |
| -w/o Thought Reward | 13.74 | 4.72 | 20.41 |
| -w/o Exploration Reward | 11.84 | 5.21 | 21.19 |
| -w/o Exploration and Thought Reward | 9.33 | 3.52 | 15.27 |

## A.9 COMPUTATIONAL OVERHEAD

We compare our computation overhead against uniform reasoning (w/o difficulty-aware CoT) under the same size of training data. The difficulty definition is calculated on the base model (Qwen2.5VL-7B) assuming 10 tokens inference, and the CoT generation is computed based on Qwen2.5VL-72B assuming 30 tokens inference. As shown in Fig. 6, the average ratio of hard steps (Level 4 and 5) is 55%, which requires CoT generation in adaptive reasoning. Uniform reasoning requires CoT generation on all data. And the two methods in CoT pretraining require similar computational costs, which could be ignored. As a result, the FLOPs of each stage is presented in Tab. 8. Although our method requires 23.5% more FLOPs in total, the average accuracy is 8.79% higher.

Table 8: FLOPs of additional difficulty definition and CoT generation stage. EM refers to average action parameter match rate.

| Type of Reasoning | Difficulty Definition$(*10^{12})$ | CoT Generation$(*10^{12})$ | EM |
|---|---|---|---|
| Adaptive Reasoning | 60.58 | 48.63 | 72.30 |
| Uniform Reasoning (w/o difficulty-aware CoT) | - | 88.42 | 63.51 |

## A.10 EXAMPLES OF OUR METHOD

To visually present the results of AdaGUI-R1, we demonstrate the visualization of some trajectories in Fig. 11, 9, and 10. Each trajectory shows the user query, screenshots, the action, and the thinking process (if any). These examples demonstrate the effectiveness of difficulty-aware reasoning of AdaGUI-R1: easy steps with few thinking, hard steps with helpful reasoning.

## A.11 QUALITATIVE COMPARISONS

To demonstrate the effectiveness of our method more intuitively, we present how our approach not only improves the accuracy of hard steps (Fig. 14), but also avoids errors caused by the hallucination triggered by over analysis in easy steps (Fig. 12 and 13). Compared with uniform-reasoning strategy (w/o difficulty-aware CoT), our method shows better performance.

To demonstrate the impact of adaptive thought reward on thought content, we show an example in Fig. 15 for a more intuitive understanding

**User query:** Add "The Road Less Traveled" book by Elizabeth Billingsley to my Flipkart app cart.

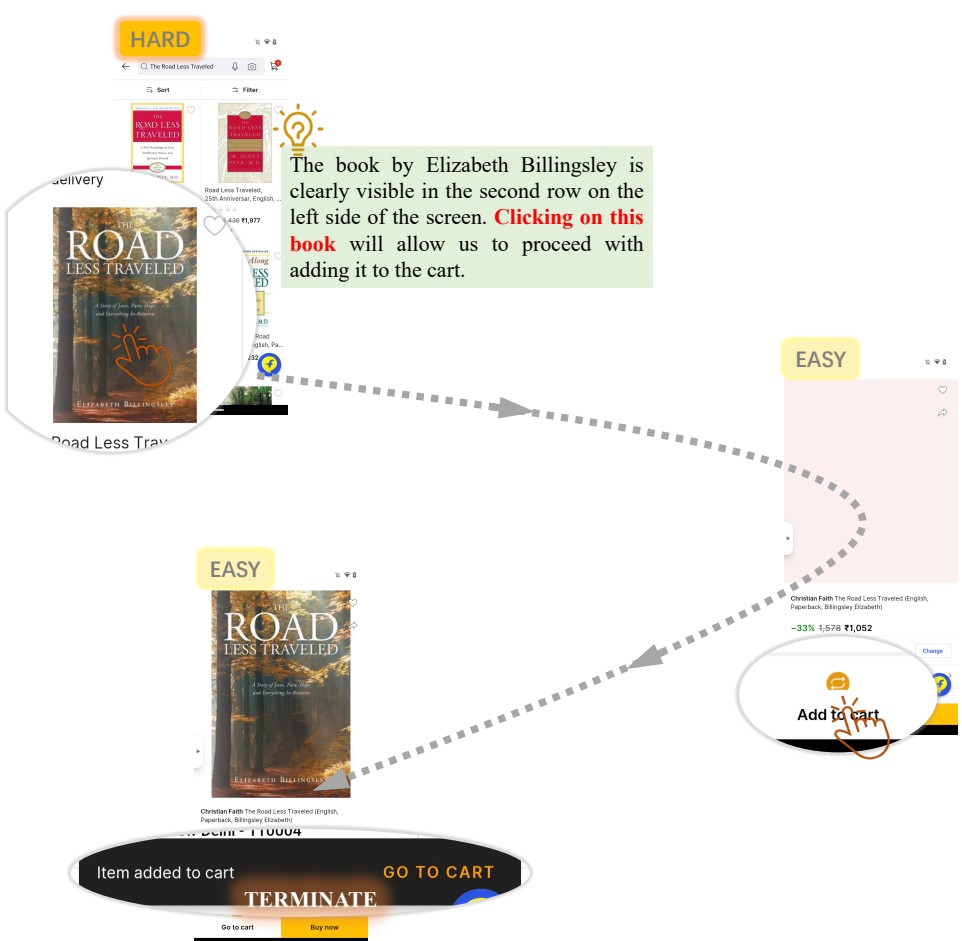

The book by Elizabeth Billingsley is clearly visible in the second row on the left side of the screen. **Clicking on this book** will allow us to proceed with adding it to the cart.

Figure 9: Example of our method.

**User query:** In the Tata Cliq app, look for the Shoes with the EVA sole.

To find shoes with EVA sole material, I need to access the 'Sole Material' filter. **Clicking on 'Sole Material'** is necessary to proceed with this filter.

To proceed, the user should **select the "EVA" option** under the "Sole Material" section.

Figure 10: Example of our method.

**User query:** Share all the details of moon phase to my wife at clarawagner98b@gmail.com via Gmail app from Phases of the Moon app.

I need to return to the main menu to access the moon phase information. **Pressing the back button** will help me navigate away from this screen.

clarawagner98b@gmail.com

The email address has been entered correctly. The send button is typically represented by an arrow icon, which is visible in the top right corner of the screen. **Clicking this button** will complete the task of sending the email.

Figure 11: Example of our method.

**User query:** In the pinterest app, I would like to watch video of rose flower drawing.

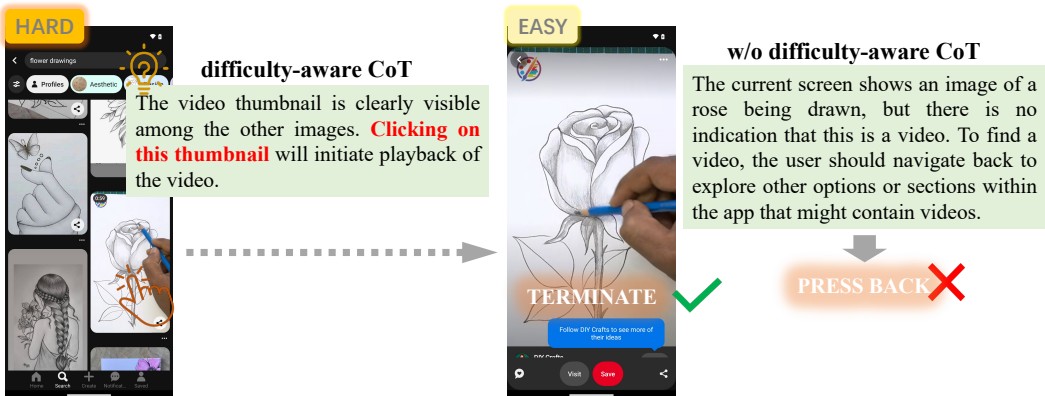

**difficulty-aware CoT**

The video thumbnail is clearly visible among the other images. **Clicking on this thumbnail** will initiate playback of the video.

**w/o difficulty-aware CoT**

The current screen shows an image of a rose being drawn, but there is no indication that this is a video. To find a video, the user should navigate back to explore other options or sections within the app that might contain videos.

Figure 12: Comparison with uniform-reasoning strategy. Our method avoids errors caused by the hallucination triggered by over analysis in easy step compared with method w/o difficulty-aware CoT.

**User query:** Show me the today workout plan.

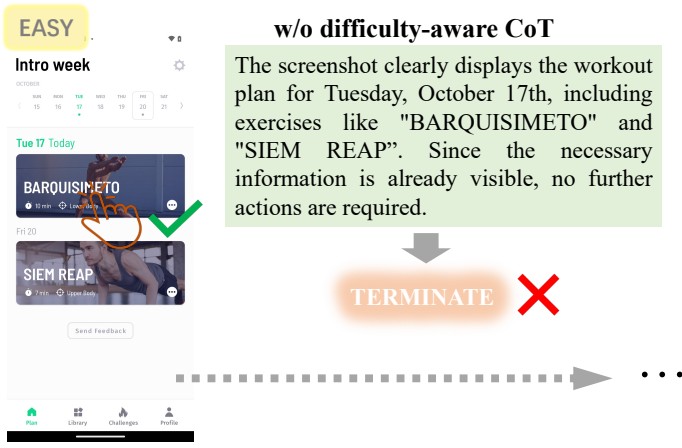

**w/o difficulty-aware CoT**

The screenshot clearly displays the workout plan for Tuesday, October 17th, including exercises like "BARQUISIMETO" and "SIEM REAP". Since the necessary information is already visible, no further actions are required.

Figure 13: Comparison with uniform-reasoning strategy. Our method avoids errors caused by the hallucination triggered by over analysis in easy step compared with method w/o difficulty-aware CoT.

**User query:** To stay informed about the state of affairs in my area, I would like to read the local news from Knoxville on the Fox news app.

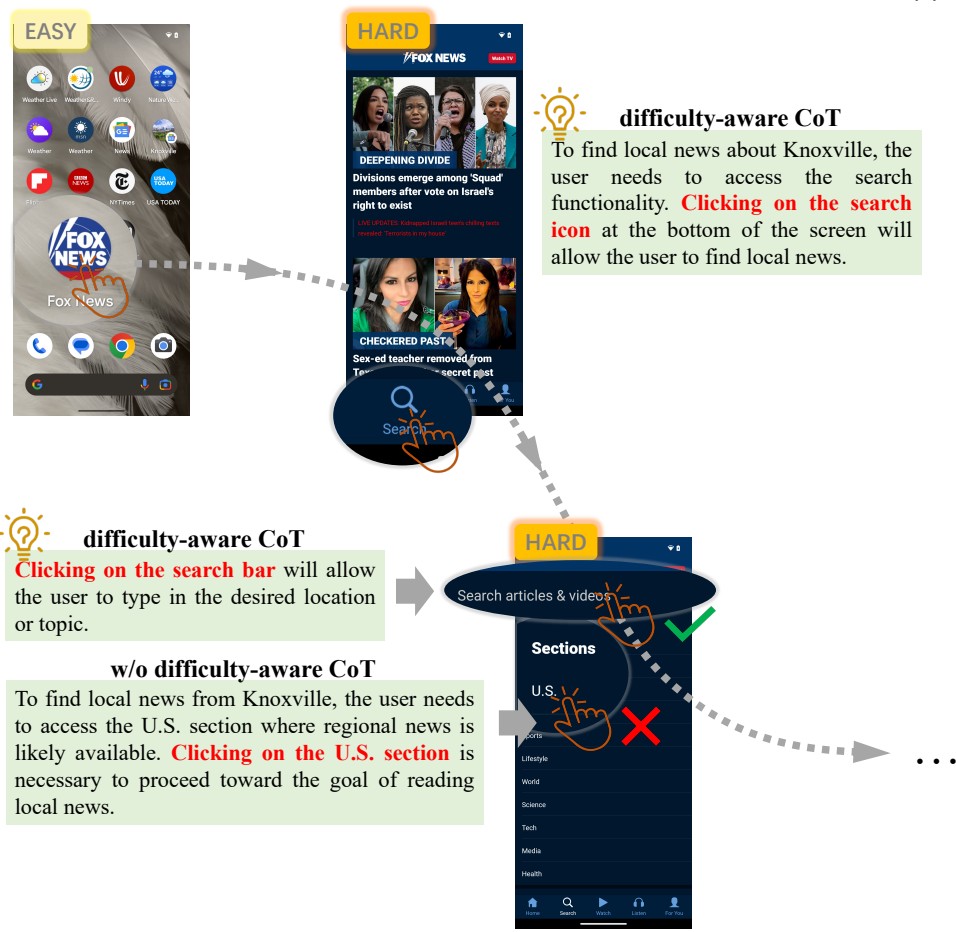

Figure 14: Comparison with uniform-reasoning strategy. Our method using difficulty-aware CoT performs correctly in hard step compared with method w/o difficulty-aware CoT.

**User query:** share the meeting reminder with dbwscratch.test.id9@gmail.com.

**HARD**

**w/ adaptive thought reward**
The meeting is visible on the screen, and **clicking on it** will likely open the detailed view where sharing options are available.

**w/o adaptive thought reward**
The meeting is displayed on-screen. Clicking on it will launch the detailed view, which contains available sharing options.

**EASY**

**w/o adaptive thought reward**
Click the button in the upper right corner of the pop-up window to perform sharing.

**EASY**

dbwscratch.test.id9@gmail.com

**w/o adaptive thought reward**
Type email address dbwscratch.test.id9@gmail.com in the recipient input box.

**HARD**

**w/ adaptive thought reward**
The send button is typically represented by an arrow icon, which is visible in the top right corner of the screen. **Clicking this button** will complete the task of sharing the meeting reminder.

**w/o adaptive thought reward**
The send button is represented by an arrow icon, visible in the top right corner of the screen. Clicking this button will complete the task of sharing the meeting reminder.

**EASY**

**w/o adaptive thought reward**
The task has been completed.

Figure 15: Comparison with no adaptive thought reward. Our method reduce the number of reasoning tokens compared with method w/o adaptive thought reward.

