# OpenReview forum: "Difficulty-Aware Reasoning for Mobile GUI Automation via Reinforcement Fine-Tuning"
_ICLR.cc/2026/Conference — Submitted to ICLR 2026_

### Official Review · Reviewer_issC · 2025-10-23

**Soundness:** 3
**Presentation:** 3
**Contribution:** 2
**Rating:** 4
**Confidence:** 5

**Summary:**

The paper *“Difficulty-Aware Reasoning for Mobile GUI Automation via Reinforcement Fine-Tuning (AdaGUI-R1)”* introduces a new framework for mobile GUI agents that can dynamically adjust how much reasoning they perform based on the difficulty of each step. Traditional GUI automation models apply uniform reasoning chains to all tasks, which leads to inefficiency—simple steps are overanalyzed while complex ones lack adequate reasoning. AdaGUI-R1 addresses this by integrating **difficulty-aware reasoning** into both supervised and reinforcement fine-tuning stages.

In the first stage, the model learns when to think by generating “Think–Action” pairs only for difficult steps, while easy ones receive a placeholder “None” thought. A self-supervised consistency mechanism ensures that generated reasoning aligns with the correct actions. In the second stage, the authors propose Group Adaptive Policy Optimization (GAPO), which introduces two key rewards: an adaptive thought reward that encourages longer reasoning for hard steps and shorter for easy ones, and a Gaussian exploration reward that provides smoother, distance-based feedback for click actions with difficulty-sensitive variance.

Experimental results on multiple mobile GUI benchmarks show that AdaGUI-R1 outperforms prior state-of-the-art methods, achieving around **5% higher action accuracy** while reducing reasoning token usage by **about 40%**. The study demonstrates that allocating reasoning effort adaptively—thinking deeply only when needed—can improve both efficiency and robustness in GUI automation agents.

**Strengths:**

The paper contributes a new difficulty-aware reasoning paradigm for GUI automation. Instead of applying a uniform Chain-of-Thought across all tasks, it introduces a principled way to adjust reasoning depth based on estimated step difficulty. This rethinking of how reasoning effort should be distributed marks a clear conceptual advancement over prior “one-size-fits-all” reasoning frameworks. The work provides several concrete and novel algorithmic components:
   - A self-supervised CoT generation mechanism that ensures consistency between thought and action.
   - The Group Adaptive Policy Optimization (GAPO) algorithm, integrating adaptive thought rewards and Gaussian exploration rewards.

   Together, these elements enhance stability, exploration efficiency, and reasoning adaptability, forming a cohesive and technically sound framework.

AdaGUI-R1 achieves substantial performance improvements on multiple GUI automation benchmarks, increasing success rates while reducing reasoning token usage by about 40%. These empirical gains demonstrate that adaptive reasoning not only improves efficiency but also sets a foundation for broader applications in multimodal and interactive AI systems.

**Weaknesses:**

1. The paper primarily evaluates the model on multiple offline benchmarks, where the metrics focus on step-level accuracy. However, these benchmarks differ from the more commonly used online interactive benchmarks (e.g., AndroidWorld, AndroidLab) that measure full-task success rates (SR). It is recommended to include a discussion on how the proposed method relates to these online benchmarks, and to report additional results of AdaGUI-R1-7B and its ablation models on such interactive benchmarks. Demonstrating effectiveness on SR metrics would significantly strengthen the paper’s empirical validity.

2. This paper emphasizes step-level difficulty awareness, with extensive design innovations in the “think” component compared to prior work. It would be beneficial to provide several case analyses, including examples of how steps are categorized by difficulty, and how the thought content changes before and after training with the Thought Reward mechanism. Such qualitative insights would clarify the behavioral impact of the proposed difficulty-aware design.

3. The Action Exploration Reward section improves upon the conventional binary (0/1) feedback by introducing a Gaussian Exploration Reward, which smooths the reward function for individual actions. However, computing Gaussian functions introduces additional computational cost compared to previous approaches. It is recommended to provide comparative experiments—such as evaluating against baselines that use bounding-box inclusion or distance-based penalty smoothing—to demonstrate that this extra computational overhead yields meaningful performance benefits.

**Questions:**

The questions are already included within the weaknesses section.

---

> ### Author Response · Authors · 2025-11-21
> **Response to Reviewer issC**
>
> Dear Reviewer issC:
>
> Thank you for your feedback and valuable suggestion! We sincerely appreciate the time and effort you have dedicated to reviewing our work. Below, we meticulously provide responses to each of your comments and outline the modifications made to the manuscript. All revisions are highlighted in red.
>
> ------------
>
> ### *W1: The offline benchmarks differ from the more commonly used online interactive benchmarks that measure full-task success rates (SR). It is recommended to include a discussion on how the proposed method relates to these online benchmarks.*
>
> The offline benchmarks is also widely used, which is sufficient to demonstrate the effectiveness of our method. Also, these offline benchmarks also contain full task success rates, which refers to the successful prediction of each step in the trajectory. The full task success rate of our full and ablation models are listed as below:
>
> | Model | GUI Odyssey | AITZ | Android Control |
> |--- | :---: | :---: | :---:|
> | **AdaGUI-R1-7B (full)** | **15.62** | **7.91** | **25.12** |
> | -w/o self-supervised CoT | 10.91 | 3.01 | 16.27 |
> | -w/o difficulty-aware CoT | 9.42 | 3.26 | 14.61 |
> | -w/o Thought Reward | 13.74 | 4.72 | 20.41 |
> | -w/o Exploration Reward | 11.84 | 5.21 | 21.19 |
> | -w/o Exploration and Thought Reward | 9.33 | 3.52 | 15.27 |
>
> Our method has not been optimized for online scenarios, including considering the complexity of online interactions and data construction. We will conduct corresponding research and improvement on online benchmarks in the future.
>
> ------------
>
> ### *W2: Qualitative insights would clarify the behavioral impact of the proposed difficulty-aware design.*
>
> We supplement some trajectories in Appendix Fig. 9, 10, and 11, where the difficulty of each step is highlighed. Also, the qualitative result of using thought reward or not is additonally shown in Fig. 15.
>
> ------------
>
> ### *W3: Provide comparative experiments on action exploration reward.*
>
> We have conducted experiments by comparing our gaussian exploration reward and box exploration reward. The results on Android Control benchmark are shown as follow:
>
> | Model |EM | Δ$_{EM}$ | Tok. | $Ratio_{Tok.}$ |
> |-------|:---:|:-----:|:------:|:----:|
> | Gaussian action reward | 72.40 | - | 13.19 | - |
> | box action reward | 69.93 | -3.41% | 13.27 | 1.01× |
>
> It could be indicating that the smooth, distance-weighted signal of gaussian exploration reward helps the policy escape local maxima, justifying its adoption in AdaGUI-R1.
>
> ------------
>
> Once again, we deeply appreciate your thoughtful and encouraging feedback. Your suggestions have enhanced the current work.
>
> Best,
>
> Authors of Paper 6643

---

> > ### Comment · Reviewer_issC · 2025-11-25
> >
> > Thank the authors for their response and detailed clarifications.
> > However, the current reply does not address my main concern. The reason for testing on online benchmarks is that, in online benchmarks, the agent is evaluated solely based on whether it ultimately fulfills the task requirements. This evaluation metric aligns with the agent’s actual utility, rather than relying on step-level accuracy to approximate real task completion. In recent prominent works, online benchmarks have been the primary evaluation focus (although these benchmarks also include offline tests), such as UI-TARS, JT-GUIAgent, and MobileUse. All of these works were published earlier this year.
> >
> > Meanwhile, the full task success rate provided by the authors is not impressive. For the full version of Android Control (which I understand to be AndroidControl-High—is that correct?), UI-TARS-7B reports an SR of 72.5, whereas the authors’ reported score is only 25.12, with the other full task success rates even lower. This also suggests that the model trained using the method described in this paper is not suitable for use in an online environment.
> >
> > In addition, I have a question regarding the comparison results:
> > The Android Control scores reported in the UI-TARS paper (https://arxiv.org/pdf/2501.12326
> > ) (Table 8, TM=98, Grounding=89.3) appear to be higher than those reported in your Table 1. Why is this the case?

---

> ### Author Response · Authors · 2025-11-26
> **Clarification on Success Rate (SR) Metrics**
>
> Dear Reviewer issC:
>
> Thank you for your follow-up and for providing these detailed and insightful comments. We  appreciate you bringing this issue to our attention. Your rigorous review has helped us identify a critical ambiguity in existing literature and a genuine error in our table.We will address your points in order:
>
> ### 1.Clarification on Success Rate (SR) Metrics
> Your primary concern stems from the comparison between the SR of 72.5% reported by UI-TARS and our SR of 25.12%. The root cause of this discrepancy is a fundamental difference in the definition of the "SR" metric used in the two contexts.
> #### 1.1 Step-level Success Rate (Step SR) in UI-TARS:
> As described in Table 7 of the UI-TARS paper, their SR metric measures accuracy at the step level. In their Table 8, this metric is not explicitly defined, leading to potential misinterpretation. The "SR" in UI-TARS is essentially equivalent to what we and other works (e.g., AgentCPM, Table 3, https://arxiv.org/pdf/2506.01391) refer to as Exact Match (EM) step accuracy. It is calculated as:
> \begin{equation}
>     EM=\frac{\sum_{i=1}^{N}  {\sum_{j=1}^{L_i}} \mathbb{1} (\hat{a}\_{ij}= {a}\_{ij}) }{M}
> \end{equation}
> where $N$ is the total number of trajectories, $L_i$ is the length of the $i$-th trajectory,  $M=\sum_{i=1}^{N}L_{i}$ is the total number of steps. This is equivalent to num_successful_steps / total_steps.
>
> #### 1.2  Full Task/Trajectory Success Rate:
> We use a much stricter definition, where a task is only considered successful if every single step in the entire trajectory is correct. The formula is:
>
> \begin{equation}
>     SR = \frac{\sum_{i=1}^{N} {\prod _{j=1}^{L_i}} \mathbb{1} (\hat{a}\_{ij}= {a}\_{ij}) }{N}
> \end{equation}
>
> This is equivalent to num_successful_trajectories / total_trajectories.
>  Therefore, a direct comparison between 72.5% (Step SR) and 25.12% (Full Task SR) is not methodologically valid.The correct comparison should be made on step-level metrics.
>
> ### 2. Corrected Comparison and Discrepancy in Table 1
> You are absolutely correct to point out the discrepancy in our Table 1. We sincerely apologize for this error.
> - Error Acknowledgment: During the table creation, we mistakenly copy-pasted the scores for Qwen2.5VL-7B-base into the row for UI-TARS. Thank you for catching this.
> - Corrected Results: We have re-verified and corrected the step-level performance of UI-TARS on Android Control, referencing the original paper, the official AgentCPM leaderboard（https://github.com/OpenBMB/AgentCPM-GUI/tree/main/eval）, and our own reproduction.
> The corrected values are:
> | Model (Source)  | TM (%) | EM (%) |
> |-------------------|:---:|:---:|
> | UI-TARS-7B(Our Incorrect Table) | 75.10  | 62.90  |
> | UI-TARS-7B (Official Paper)      | 83.7   | 72.4  |
> | UI-TARS-7B (AgentCPM Report)     | 81.6   | 74.4   |
> | UI-TARS-7B (Our Reproduction)    | 81.6   | 74.45  |
>
> ### 3.Regarding Online vs. Offline Benchmarks
> We completely agree with you on the importance of online benchmarks as the ultimate test of an agent's real-world utility. Our decision to focus on offline evaluation was driven by the following:
> - Offline, step-by-step evaluation provides a controlled and reproducible environment to analyze an agent's core reasoning and action prediction capabilities. It allows us to precisely diagnose where and why a model fails, which is crucial for advancing the underlying science.
> - As seen in prominent works like AgentCPM, InfiGUI-R1 and Aguvis , detailed offline analysis remains a standard and vital part of agent evaluation.
>
> We will revise our paper to make these distinctions crystal clear and explicitly state the definitions of all metrics used. Thank you once again for your invaluable feedback, which has significantly helped us improve the clarity and accuracy of our paper.
>
> Best,
> Authors of Paper 6643

---

### Official Review · Reviewer_HLM6 · 2025-10-26

**Soundness:** 3
**Presentation:** 3
**Contribution:** 2
**Rating:** 4
**Confidence:** 4

**Summary:**

This paper presents AdaGUI-R1, a mobile GUI agent that introduces difficulty-aware reasoning, dynamically adjusting its reasoning depth based on task complexity. The method integrates a self-supervised CoT generation process to produce consistent reasoning-action pairs and a Group Adaptive Policy Optimization (GAPO) algorithm with adaptive thought and exploration rewards. Experiments on multiple GUI automation benchmarks demonstrate significant improvements in both accuracy and efficiency.

**Strengths:**

- Innovative reward design: The paper introduces a difficulty-aware reward mechanism that assigns different reward functions to actions of varying difficulty, effectively aligning reasoning depth with task complexity.

- Strong empirical results: The proposed AdaGUI-R1 achieves strong performance on three benchmarks, surpassing prior models in both accuracy and efficiency.

**Weaknesses:**

- More qualitative examples (e.g., reasoning traces comparing easy vs. hard steps) are required.
- Lack of novelty in "Self-Supervised CoT Generation": The proposed "self-supervised CoT generation" closely mirrors the STaR [1] approach and does not introduce a fundamentally new mechanism. The pipeline of generating initial CoT, validating actions, and revising reasoning is nearly identical to prior methods.
- Soft reward on coordinates is not new: Using spatially smoothed rewards for click or grounding actions has been explored in previous RL-based GUI grounding works.

[1] STaR: Bootstrapping Reasoning With Reasoning.

**Questions:**

- What is the performance of model after SFT.
- According to the experimental results, the average reasoning token length of AdaGUI-R1 is less than 20 tokens, which suggests that the model performs almost no explicit reasoning. So does it mean that it is enough to train the model to predict actions? And have you observed that the decrease in thought length is mainly due to the omission of content?
- What content or reasoning elements are being omitted in the reduction of cot length?

---

> ### Author Response · Authors · 2025-11-21
> **Response to Reviewer HLM6**
>
> Dear Reviewer HLM6:
>
> Thank you for your feedback and valuable suggestion! We sincerely appreciate the time and effort you have dedicated to reviewing our work. Below, we meticulously provide responses to each of your comments and outline the modifications made to the manuscript. All revisions are highlighted in red.
>
> ------------
>
> ### *W1: More qualitative examples are required.*
>
> We supplement some examples in Appendix Sec. A.10, as shown in Fig. 9, 10, 11.
>
> ------------
>
> ### *W2: The proposed "self-supervised CoT generation" closely mirrors the "STaR: Bootstrapping Reasoning With Reasoning." approach.*
>
> Our "Self Superior CoT Generation" method is different from STaR. Their approach involves adding few shot examples to the prompt to guide the model in generating the corresponding CoT, and then verifying the correctness of the predicted action. **When the action is correct**, the think-action pair is applied as a new learning sample. **When the action is incorrect**, the model is prompted with the ground truth action in the prompt to obtain the CoT.
>
> Our method differs from theirs in two ways:
> - **We do not include few shot examples** in the prompt to guide the model in generating the corresponding CoT.
> - After obtaining the preliminary CoT through prompting the model ground truth action, **we also add a validation process for the CoT**, which injects the CoT into the prompt for the model to predict the action. **By comparing the predicted action with the ground truth action, we judge the quality of the CoT.** If the match fails, we prompt the model ground truth action to be different from the predicted action, and ask it to modify the previously generated CoT to obtain an refined CoT. For correctly matched examples, we directly use think-action pairs as data samples.
>
> ------------
>
> ### *W3: Soft reward on coordinates is not new.*
>
> In the design of action exploration rewards, we not only consider **the impact of discrete and continuous rewards** on click operations, but also **further explore the addition of difficulty scales in continuous rewards**. Introducing different Gaussian bandwidths according to the difficulty of different steps helps improve the operational precision of the model in the RL stage.
>
> ------------
>
> ### *Q1: What is the performance of model after SFT?*
>
> The performance of EM metric (action parameter match rate) after SFT is shown in the table below:
>
> | Model | GUI Odyssey | AITZ | Android Control |
> |------|:-----:|:----:|:-----:|
> | Qwen2.5-VL-7B-Base | 60.58 | 48.63 | 72.30 |
> | Stage 1: SFT | 77.62 | 65.33 | 71.86 |
> | Stage 2: RFT | 77.88 | 66.49 | 72.82 |
>
> ------------
>
> ### *Q2 & Q3: The average reasoning token length of AdaGUI-R1 is less than 20 tokens, which suggests that the model performs almost no explicit reasoning. Have you observed that the decrease in thought length is mainly due to the omission of content? What content or reasoning elements are being omitted in the reduction of cot length?*
>
> An average thinking length of less than 20 tokens does not mean that the model has almost no explicit thinking. As shown in Figure 7 in the appendix, after training, the proportion of hard steps in the model is about 40%, while the proportion of easy steps is about 60%. The average thinking length of hard steps is about 35 tokens, while easy steps directly output empty `<think></think>` with a length of 0. The weighted average reasoning token is 0.4 × 35 + 0.6 × 0 ≈ 14 tokens. **Our model does not skip reasoning, but generates a complete CoT only when it is truly needed.** The explicit reasoning process is still necessary for hard steps and has not been omitted. For easy steps, the model learns through training to express enough signal with minimal tokens. The decrease in thought length belongs to information compression rather than information dropout.
>
> ------------
>
> Once again, we deeply appreciate your thoughtful and encouraging feedback. Your suggestions have enhanced the current work.
>
> Best,
>
> Authors of Paper 6643

---

### Official Review · Reviewer_GsMa · 2025-10-31

**Soundness:** 2
**Presentation:** 2
**Contribution:** 1
**Rating:** 2
**Confidence:** 3

**Summary:**

Automating GUI tasks is challenging because of the task complexity. State-of-the-art employs chain-of-thought reasoning in order to deal with such complexity; however, they equally apply the reasoning protocol without considering the difficulty or complexity of the sub-tasks. This results in computational inefficiency because of applying unnecessary reasoning steps for trivial sub-tasks. Also, such a one-fit-for-all approach results in performance degradation, especially in complex subtasks, since a fixed number of reasoning steps may not be enough for some complex sub-tasks, while being more than enough for trivial ones. This paper introduces a difficulty-aware reasoning, which adapts the depth of reasoning to the action complexity. The core idea is determining the difficulty of subtasks via a pre-trained VLM for GUI tasks. Then, a reward component is employed to reward the model for thinking longer for harder tasks and shorter for easier ones. The proposed method shows performance improvements over baselines for GUI tasks.

**Strengths:**

- **Difficulty-aware Reasoning**: Determining the difficulty and encouraging the agent to think longer for harder sub-tasks for GUIs looks promising for GUI agents.
- **Improvements over Baselines**: Results are promising, showing significant improvements over the base model they compared.
- **Reward Components**: Reward components are ablated nicely, and it is shown that each component helps the agent to reach a better performance.
- **Difficulty-threshold Analysis**: The effects of the difficulty threshold, which takes a key role in $R_{thought}$, on the model performance is analyzed nicely.

**Weaknesses:**

**Major**:
- **Novelty**: The paper's claims and coverage are scoped entirely to GUI automation, and the key contribution described as the difficulty-aware CoT. However, there are already works in literature where CoT is adapted based on the task difficulty (see below). These works must be discussed in detail, and the proposed approach should be compared with them, since the novelty of the proposed method is questionable beyond the experimental setting.

[a]: Waheed, Abdul, et al. "Less is More Tokens: Efficient Math Reasoning via Difficulty-Aware Chain-of-Thought Distillation." arXiv preprint arXiv:2509.05226 (2025).

[b]: Yu, Zishun, et al. "Think Smarter not Harder: Adaptive Reasoning with Inference Aware Optimization." Forty-second International Conference on Machine Learning.

[c]: Wang, Xinglin, et al. "Make Every Penny Count: Difficulty-Adaptive Self-Consistency for Cost-Efficient Reasoning." Findings of the Association for Computational Linguistics: NAACL 2025. 2025.

[d]: Han, Tingxu, et al. "Token-budget-aware llm reasoning." arXiv preprint arXiv:2412.18547 (2024).

[e]: Aggarwal, Pranjal, Aman Madaan, and Yiming Yang. "Let’s Sample Step by Step: Adaptive-Consistency for Efficient Reasoning and Coding with LLMs." Proceedings of the 2023 Conference on Empirical Methods in Natural Language Processing. 2023.

[f]: Damani, Mehul, et al. "Learning How Hard to Think: Input-Adaptive Allocation of LM Computation." The Thirteenth International Conference on Learning Representations.

- **Computational Overhead**: The proposed approach requires difficulty estimation and CoT pre-training. However, the computational overhead over the baselines are not discussed. This must be clearly elaborated, and equal-compute comparisons and a quality–cost curve (tokens/FLOPs/wall-clock) should be presented.
- **Discrete Difficulty Levels**: It is unclear why only five discrete difficulty levels are selected. This design choice is not explained.
- **CoT Pre-training**: It is mentioned that they first teach how to generate CoT to their agent, using a curated, annotated dataset. The details of such fine-tuning and the data size are unclear.
- **Reward Design**: It is unclear how R_{thought} is designed.  It would also be great to show the impact of this reward design on the model's performance by comparing it with simpler designs.
- **Experimental Results**: In the experimental results in Table 1, for AITZ, UI-TARS-7B works better than the proposed approach. However, these results are not written in bold; instead, the results of the proposed approach are written in bold, which is misleading. This must be corrected. Also, please elaborate on why UI-TARS-7B performs better than the proposed approach.
- **Confidence Intervals**: No confidence intervals are presented for the results in Table 1. Please report the confidence intervals either in the main table or in the appendix. This is important since it looks like UI-TARS-7B performs closely to the proposed approach.

**Minor**:
- **Presentation**: Figure 1-right is very unclear. Please either explain what this figure shows in the caption in detail or remove it.

**Questions:**

- In figure 1-right, why do we have multiple bars for different levels? What are these levels, the difficulty levels according to Eq. 4?
- Why are there five discrete difficulty levels? Have you ever considered continuous difficulty levels, or are there any drawbacks to using them?
- Please explain how you teach "how" and "when" to generate CoT to your agent. Can you also elaborate on the data curation and the size of the data used for such pre-training?
- How is R_{thought} designed? It is mentioned that the function is smooth and strictly monotonic, but is this the only design consideration? Have you ever considered different functions?
- Why does UI-TARS-7B perform better than the proposed approach in AITZ? This must be clearly elaborated.
- Can you please report confidence intervals for the results in Table 1?

---

> ### Author Response · Authors · 2025-11-21
> **Response to Reviewer GsMa (1/2)**
>
> Dear Reviewer GsMa:
>
> Thank you for your feedback and valuable suggestion! We sincerely appreciate the time and effort you have dedicated to reviewing our work. Below, we meticulously provide responses to each of your comments and outline the modifications made to the manuscript. All revisions are highlighted in red.
>
> ------------
>
> ### *W1: There are already works where CoT is adapted based on the task difficulty. The proposed approach should be compared with them.*
>
> Our difficulty-aware and adaptive CoT method differs from previous methods in the following aspects:
> 1. In terms of difficulty definition, some previous methods ([a], [b]) rely on the inference of closed-source models or existing definitions in the dataset, but **we determine the difficulty level based on the capabilities of the base model itself**. Our definition is more reasonable and focuses on improving the thinking ability of the base model.
> 2. The previous adaptive reasoning was based on LLM methods, while our method requires processing both image and text inputs simultaneously. Some methods ([b], [c], [d], [e], [f]) define difficulty based on text length or logits-related factors, ignoring pixel density and spatial ambiguity of screenshots. Our difficulty of a single step comes from the combined confidence level of image and text, and **logits alone cannot determine whether the final action matches the ground truth**. Our method is more in line with real GUI interaction.
>
> Meanwhile, due to different modal inputs, it is difficult to experimentally compare with these adaptive think schemes based on LLM as they could not process image input.
>
> ------------
>
> ### *W2: The proposed approach requires difficulty estimation and CoT pre-training. The computational overhead over the baselines are not discussed.*
>
> We list the computational overhead in the table below.
>
> | Type of Reasoning | Difficulty Definition(*10¹²) | CoT Generation(*10¹²) | EM |
> |-------------------|:------------------------------:|:------------------------:|:-----:|
> | Adaptive Reasoning | 60.58 | 48.63 | 72.30 |
> | Uniform Reasoning (w/o difficulty-aware CoT) | - | 88.42 | 63.51 |
>
> We compare our computation overhead against uniform reasoning (w/o difficulty-aware CoT) under the same size of training data. The difficulty definition is calculated on the base model (Qwen2.5VL-7B) assuming 10 tokens inference, and the CoT generation is computed based on Qwen2.5VL-72B assuming 30 tokens inference. As shown in Fig. 6 in the Appendix, the average ratio of hard steps (Level 4 and 5) is $55\%$, which requires CoT generation in adaptive reasoning. Uniform reasoning requires CoT generation on all data. And the two methods in CoT pretraining require similar computational costs, which could be ignored. Although our method requires $23.5\%$ more FLOPs in total, the average accuracy is $8.79\%$ higher.
>
> We also supplement these comparison in Appendix Sec. A.9.
>
> ------------
>
> ### *W3 & Q2: It is unclear why only five discrete difficulty levels are selected.*
>
> Thanks for your concerns. This is what we did not clarify in our experimental results, and we add this clarification in Sec. 4.3 of the revised version. In the experiment, we also considered various scales of difficulty, such as scales of $2, 5, 10$ in Eq.4. The results on Android Control benchmark are as below:
>
> | Model | EM | Δ$_{EM}$ | Click ACC | Δ$_{Click-ACC}$ |
> |-------|-----|------|-----------|------------|
> | AdaGUI-R1-7B (five-tier scale) | 72.40 | - | 75.16 | - |
> | -two-tier scale | 69.20 | -3.20% | 71.84 | -3.32% |
> | -ten-tier scale | 70.16 | -2.24% | 73.70 | -1.46% |
>
> For two-tier scale, the mapping funtion is $\ell(\rho) = 2 - \lceil\rho-0.5\rceil$. For ten-tier scale, the mapping funtion is $\ell(\rho) = 11 - \lceil5\rho+0.5\rceil$.
>
> Through ablation experiments, we found that the model designed with five-tier scale ultimately demonstrated the best model ability in GAPO RL-finetuning. Therefore, we chose five discrete difficulty levels as our experimental setting.
>
> ------------
>
> ### *W4 & Q3: The details of adaptive CoT fine-tuning and the data size are unclear.*
>
> **For the details of adaptive CoT fine-tuning**, we first label each data with a difficulty tag. For hard steps, we use a self-supervised approach to construct `think-action` pairs. For easy steps, we simply left `think` blank, thus expanding the training data's answer from `action` to `think-action`. We fine-tune the model using SFT, and the training parameter settings are listed in 4.1.
>
> **For the data size we use**, our training data are selected from official training sets. The official training set of four datasets consists of 233k automation steps. In order to save computational resources, we select half of the data for constructing our training set, i.e. 116.5k samples.

---

> ### Author Response · Authors · 2025-11-21
> **Response to Reviewer GsMa (2/2)**
>
> ### *W5 & Q4: It is unclear how $R_{thought}$ is designed.*
>
> The main purpose of $R_{thought}$ is to stimulate the model's thinking and exploration space for different difficulty steps, i.e. shorter CoT for easy steps and longer for hard ones. In actual design, we found that the cosine form of thought rewards for different difficulty levels has a better stimulating effect on the model's thinking mode than the delta form of thought rewards. Therefore, $R_{thought}$ in Eq. 5 design is adopted. The comparison on Android Control benchmark is listed as follow:
>
> | Model |EM | Δ$_{EM}$ | Tok. | Ratio$_{Tok.}$ |
> |-------|:---:|:-----:|:------:|:----:|
> | cosine-like thought reward | 72.40 | - | 13.19 | - |
> | delta-like thought reward | 70.27 | -2.94% | 15.12 | 1.15× |
>
> The delta form of thought rewards is expressed as $R_{thought-delta}(t|l) = \mathbb{1}[t<T/2]\ (\text{if }l\le l_{thr})\ /\ \mathbb{1}[t>T/2]\ (\text{if }l>l_{thr})$.
>
> ------------
>
> ### *W6 & Q5: Experimental results is misleading. Please elaborate on why UI-TARS-7B performs better than the proposed approach.*
>
> Sorry for the misunderstanding, we have made corrections in our revised manuscripts.
>
> In fact, **the overall performance of UI-TARS-7B on AITZ is not better than ours**. Its TM (Type Match) metric is higher, but this metric only measures whether the predicted action types (such as click, type, scroll, etc.) of the model are consistent with the ground true values, and does not involve the accuracy of specific parameters (such as coordinates, text content). On the contrary, we are better than UI-TARS-7B in the EM (Exact Match) metric, which better reflects end-to-end correctness. This indicates that our method not only accurately determines the type, but also approaches the ground truth values in specific parameters of the action (such as click position, input text, scrolling direction, etc.), resulting in a higher task completion rate.
>
> ------------
>
> ### *W7 & Q6: Please report the confidence intervals.*
>
> Thank you for providing this insightful suggestion. We report the accuracy and its 95% confidence interval computed with the Wilson score interval over each test set. The results are listed as below:
>
> | Model          |  GUI Odyssey     |             | AITZ        |             | Android Control          |             |
> |---------------|:-------------:|:-------------:|:-------------:|:-------------:|:-------------:|:-------------:|
> |               | TM     | EM     | TM     | EM     | TM     | EM     |
> | GUI-R1-3B     | 56.29(±0.57)  | 50.10(±0.57)  | 50.55(±1.43)  | 44.35(±1.42)  | 52.14(±0.97)  | 47.95(±0.97)  |
> | InfiGUI-R1-3B | 74.50(±0.50)  | 55.02(±0.57 ) | 70.77(±1.30)  | 52.88(±1.42)  | 71.53(±0.88)  | 58.04(±0.96)  |
> | GUI-R1-7B     | 63.21(±0.55)  | 55.26(±0.57)  | 56.75(±1.41)  | 50.47(±1.43)  | 57.48(±0.96)  | 51.88(±0.97)  |
> | UI-TARS-7B    | 86.06(±0.40)  | 67.90(±0.53)  | 80.42(±1.13)  | 65.77(±1.35)  | 75.10(±0.84)  | 62.90(±0.94)  |
> | AdaGUI-R1-3B  | 87.10(±0.38)  | 72.97(±0.50)  | 75.89(±1.22)  | 59.91(±1.40)  | 81.91(±0.75)  | 69.75(±0.89)  |
> | AdaGUI-R1-7B  | 89.24(±0.35)  | 77.88(±0.47)  | 78.64(±1.17)  | 66.62(±1.34)  | 82.97(±0.73)  | 72.40(±0.87)  |
>
> As AITZ is a relatively simple dataset with an average of 7.5 steps per task and the ambiguity of user query is significantly lower compared to real-world scenarios, the performance appears numerically close for UI-TAR-7B and AdaGUI-R1-7B on AITZ. However, on the more complex dataset GUI Odyssey (with an average of 15.4 steps), AdaGUI-R1 outperformed UI-TARS-7B and the confidence intervals of them do not overlap, indicating a statistically better performace of AdaGUI-R1 on complex scenarios.
>
> We have also supplemented this table in Appendix.
>
> ------------
>
> ### *Minor Weakness: Figure 1-right is very unclear.*
>
> Sorry for the confusion. Fig.1-right shows the number of reasoning token for each step within a specific trajectory with 12 steps. The step difficulty level indicates the probability of step failure (the lower, the easier). Compared to InfiGUI-R1 using uniform-reasoning mechanism, AdaGUI-R1 reduces the number of reasoning tokens.
>
> We have also added corresponding explanations to the caption in the revised version.
>
> ------------
>
> ### *Q1: Why do we have multiple bars for different levels in Fig. 1? What are these levels?*
>
> Because the difficulty level of different steps within a trajectory varies. Fig.1 shows a trajectory with 12 steps, starting from 12 o'clock clockwise. And these levels are defined according to Eq. 4, indicating the possibility of failure during prediction.
>
> ------------
>
> Once again, we deeply appreciate your thoughtful and encouraging feedback. Your suggestions have enhanced the current work.
>
> Best,
>
> Authors of Paper 6643

---

### Official Review · Reviewer_GfcT · 2025-10-31

**Soundness:** 2
**Presentation:** 3
**Contribution:** 2
**Rating:** 4
**Confidence:** 4

**Summary:**

The paper introduces reinforcement learning fine-tuning algorithm that induces difficulty-aware reasoning. The intuition is that the model should "reason" only in difficult states.

The proposed method works in two stages:
1. Inducing reasoning using supervised fine-tuning.
2. Refining reasoning to be difficulty-aware through a novel reinforcement learning algorithm: GAPO that defines rewards based on the difficulty of the decision in that state. Precisely, during fine-tuning an agent is penalised when providing an answer to a difficult step without reasoning, or for too much "thinking" in low-difficulty states.

In the first stage the difficulty is measured by another VLLM, while in the second stage the difficulty is computed on the fly based on the group of generated samples.

The suggested method outperforms other recent grounding models with or without reasoning.

**Strengths:**

- Comprehensive set of ablations for all components of the algorithm.
- Cost analysis demonstrating efficient use of tokens (a performance / tokens 2D plot would be useful for visualisation here)
- Strong empirical results.

**Weaknesses:**

1. No direct comparison with other adaptive reasoning methods.
2. Reduced novelty: adaptive thinking has been introduced before.

**Questions:**

1. Why isn't the current solution compared to AdaptThink, AdaCoT, ThinkSwitcher... works that were correctly mentioned in Section 2?
2. When computing the difficulty level $l$ during "enhancement", rather than sampling multiple times and counting how many generations exactly match the target, wouldn't be easier to measure the log-probs of the correct answer against a threshold?
3. Do all the models in Table 1 follow the same protocol of using half of the data for training and half for testing. Is the split the same?

---

> ### Author Response · Authors · 2025-11-21
> **Response to Reviewer GfcT**
>
> Dear Reviewer GfcT:
>
> Thank you for your feedback and valuable suggestion! We sincerely appreciate the time and effort you have dedicated to reviewing our work. Below, we meticulously provide responses to each of your comments and outline the modifications made to the manuscript. All revisions are highlighted in red.
>
> ------------
>
> ### *W1&Q1: No direct comparison with other adaptive reasoning methods.*
>
> There is no adaptive reasoning method in the field of GUI automation. Our method requires simultaneous processing of image and text input. The existing adaptive reasoning method is based on LLM and design for LLM reasoning, thus our method cannot be compared with LLM-based adaptive think approaches.
>
> ------------
>
> ### *W2: Adaptive thinking has been introduced before.*
>
> The concept of adaptive thinking has a precedent in general NLP, however, in the specific scenario of GUI automation, this idea is still blank. The "adaptative" of general NLP is mostly designed based on text semantic complexity, while the core challenge of GUI tasks comes from multimodal coupling factors such as visual layout mixing, element density, and user query ambiguity. Its difficulty definition and calculation are completely different from pure text. GUI tasks needs to consider information such as screenshots and interaction history simultaneously, and the complexity indicator of traditional NLP cannot be directly transferred.
>
> ------------
>
> ### *Q2: Wouldn't it be easier to measure the log-probs of the correct answer against a threshold?*
>
> A high log-prob only indicates a high level of confidence in the model's prediction, and does not necessarily mean that the predicted action is correct. It cannot accurately measure the actual difficulty of the current step. Only by comparing the final predicted action with the ground truth action can the difficulty level of the current step be calculated.
>
> ------------
>
> ### *Q3: Do all the models in Table 1 follow the same protocol of using half of the data for training and half for testing. Is the split the same?*
>
> Sorry for our misleading statement. We did not train with half the data and test with half the data. We trained with half the data from all official training sets, and the results were calculated on the official test set. Among all the compared methods, the test data are the same. The training data used by other models varies, detailed in their articles, and we list them as below.
>
> | Model            | Train Dataset                                                                 |
> |--------|-----------|
> | Aguvis        | Amex, AITZ, Android Control, GUI Odyssey, AITW, GUIAct, MiniWoB++ |
> | OS-Atlas-Pro | Amex, AITZ, Mind2Web  |
> | OdysseyAgent  | GUI Odyssey  |
> | GUI-R1           | FineWeb, UIBert, AMEX, RICOSCA, SeeClick, OS-Otlas     |
> | InfiGUI-R1       | Android Control|
> | UI-TARS          | MM-Mind2Web, GUIAct, AITW, AITZ, Android Control, GUI Odyssey, AMEX, private data |
> | AdaGUI-R1        | Amex, AITZ, Android Control, GUI Odyssey|
>
> We have refined our statement in the revised version in Sec. 4.1, which is highlighted in red.
>
> ------------
>
> Once again, we deeply appreciate your thoughtful and encouraging feedback. Your suggestions have enhanced the current work.
>
> Best,
>
> Authors of Paper 6643

---

### Official Review · Reviewer_1NDW · 2025-11-01

**Soundness:** 2
**Presentation:** 2
**Contribution:** 2
**Rating:** 4
**Confidence:** 3

**Summary:**

The paper proposes to create a difficulty aware reasoning paradigm that aims to reduce token budget and adaptively reason longer on hard steps and avoid reasoning longer on easy ones. The paper overall manages to reduce the token budget and not reduce performance at the same time. It is a timely and interesting work  that aims to increase the metacognitive abilities of the agent by helping in its resource allocation of thinking.

**Strengths:**

The paper tackles the problem of thinking budget of the agent thereby increasing the metacognitive aspects of decision making in GUI environments. The paper also increases the efficiency of exploration of agent in a high dimensional setting as GUIs by including an action exploration reward for the hard steps where reward is sparse and making it more dense with a Gaussian exploration reward
Ablations and experiments seem robust.

**Weaknesses:**

Even though the agent is advertised as self supervised, it does require ground truth labels from existing datasets that are human annotated and in that the approach isn’t general or scalable.

There's a possibility that there is a lot of redundancy in these datasets, so I am not sure the degree of generalization that this approach promises. In general, on exploration task fine-tuning on similar trajectories improve performance. Hence results could also be explained by data leak, as the model get finetuned  on half the trajectories randomly selected. So some sort of stratified split up of the tasks in these datasets showing some analysis would be helpful to understand as in where exactly the performance is coming from.

Some figures are not referenced in the text.

**Questions:**

In line 399, the paper claims and I quote “not only improves the accuracy of hard steps, but also avoids the hallucination triggered by introducing over analysis in easy steps”, this is a serious claim but I would like to see some evidence from some analysis which you could do to show it.

There are eight actions: key, click, swipe, long press, type, system button, terminate, and wait. Is the gaussian exploration function helpful for all these actions or is it limited to some?

Any reason why this algorithm was tested only on mobile GUI environments only (not a demerit of the paper)?

Any reason you went for an easy/hard dissociation instead of a graded difficulty thinking budget. What I mean is it could have easily been a continuous function ( like agreement of high accuracy outputs) of difficulty adaptation instead of just these two distinct categories.

How do you ensure that the agent doesn't reward hack the length bonus and increases the token budget unnecessarily for hard problems which such a long length might not be needed. It could be that you are managing to reduce the token budget for easy steps but increasing it for harder ones unnecessarily with your design choices.

I would be willing to revise my scores if you could answer these questions satisfactorily and address weakness.Thank you.

---

> ### Author Response · Authors · 2025-11-21
> **Response to Reviewer 1NDW （1/2）**
>
> Dear Reviewer 1NDW:
>
> Thank you for your feedback and valuable suggestion! We sincerely appreciate the time and effort you have dedicated to reviewing our work. Below, we meticulously provide responses to each of your comments and outline the modifications made to the manuscript. All revisions are highlighted in red.
>
> ------------
>
> ### *W1: The agent require ground truth labels that are human annotated and in that the approach isn’t general or scalable.*
>
> The label requiring human annotation in the data is the ground truth `action` of the current step, which is originally included in each GUI automation dataset. CoT generation extends the original answer from `action` to` think-action` pair, where `think` in our method is generated through **self-supervised mechanism**. Thus, our self-supervised CoT generation mechanism is universal and extensible.
>
> ------------
>
> ### *W2: Results could also be explained by data leak, as the model get finetuned on half the trajectories randomly selected.*
>
> The reason for the performance improvement of our method is not due to data leak, as **we did not include test data during training**. We only use half of the data from four official training sets for training in order to save computing resources, and the results were inferred on the official test set. The test data are the same as previous researches, except that we did not use full training data for training.
>
> In addition, we also perform unified t-SNE visualization on the official training and test sets (see Appendix Fig. 8 for the result). The figure shows that the two highly overlap in 2D space, indicating similar distributions, which helps the model generalize, but does not mean there is data leak, as all screenshots and instructions are official independent partitions.
>
> ------------
>
> ### *W3: Some figures are not referenced in the text.*
>
> Thanks for pointing out the oversight that some figures are not referenced in the main text. We deeply apologize for this and have checked and supplemented the citation positions of all figure numbers one by one in the revised manuscript to ensure that the text corresponds one-to-one with the figures. The specific modification is as follows:
> - Fig.4 (Gaussian exploration reward) $\rightarrow$ refer to the last paragraph Sec. 3.4.2 for new reference
>
> All figures have now been clearly cited in the main text and are uniformly presented in the "Fig. X" format to avoid the recurrence of isolated figures. Thank you again for your careful review and reminder.

---

> ### Author Response · Authors · 2025-11-21
> **Response to Reviewer 1NDW (2/2)**
>
> ### *Q1:  Claims of “not only improves the accuracy of hard steps, but also avoids the hallucination triggered by introducing over analysis in easy steps”*
>
> We additionally present some examples in the Appendix. Fig. 14 shows that our approach improves the accuracy of hard steps, and Fig. 12 and 13 demonstrate that our method avoids errors caused the hallucination triggered by introducing over analysis in easy steps.
>
> Hope that these updates could meet your expectations.
>
> ------------
>
> ### *Q2: Is the gaussian exploration function helpful for all these actions or is it limited to some?*
>
> The first paragraph in Sec. 3.4.2 (L340-342) explains that our Gaussian exploration reward is targeted at `Click` and is not helpful for other actions. As shown in Tab. 4 in the Appendix, the proportion of `Click` is the highest in all datasets, so we mainly focus on improving the accuracy of `Click` using Gaussian exploration reward.
>
> ------------
>
> ### *Q3: Why this algorithm was tested only on mobile GUI environments only*
>
> In order to quickly validate the algorithm, we choose mobile GUI environment with limited computing resources, as the resolution of a screenshot on mobile is only $\frac{1}{2}$~$\frac{2}{3}$ of on web/desktop (typically 1179×2556 vs 2560×1664). In fact, our method itself is not bound to the platform and will be extended to desktop/web scenarios in the future.
>
> ------------
>
> ### *Q4: Why choose an easy/hard dissociation instead of a graded difficulty thinking budget?*
>
> In order to match the classification of easy and hard steps during inducing reasoning paradigm, we use the same of two distinct categories when designing adaptive thought reward, and design two functions separately. In fact, using a continuous function based on accuracy score is also possible. We have conducted additional experiments and the results on Android Control benchmark are as follows:
>
> | Model |EM | Δ$_{EM}$ | Tok. | $Ratio_{Tok.}$ |
> |-------|:---:|:-----:|:------:|:----:|
> | easy/hard dissociation thought reward | 72.40 | - | 13.19 | - |
> | graded thought reward | 72.03 | -0.51% | 16.92 | 1.28× |
>
> The continuous function is expressed as $R_{\text{thought-acc-based}}(t, \rho) = (1-\rho) * (1 - e^{-t/T}) + \rho * e^{-t/T}$. Our results are better because the continuous budget continuous accuracy-based thought reward causes fuzzy signals for mid-difficulty steps ($\rho \in [0.4, 0.6] $), resulting in increase of reasoning token. Our two-category method effectively eliminates this "grey-zone" by applying a easy/hard dissociation.
>
> ------------
>
> ### *Q5: How do you ensure that the agent doesn't reward hack the length bonus and increases the token budget unnecessarily for hard problems?*
>
> Our goal is to encourage models to think in hard steps, think less/not in easy steps, and for hard steps, RL reward=(task success+length discount) - length penalty. If the predicted action is wrong, regardless of how long the CoT is written, the length reward coefficient is directly set to 0, so simply stacking tokens cannot bring positive returns. The experimental results also showed that the overall token quantity decreased by 40% compared to the uniform inference baseline, indicating that the design not only lowered the easy step but also did not unnecessarily inflate the hard step.
>
> ------------
>
> Once again, we deeply appreciate your thoughtful and encouraging feedback. Your suggestions have enhanced the current work.
>
> Best,
>
> Authors of Paper 6643

---

### Comment · Area_Chair_7r1e · 2025-11-25

Dear Reviewers,

A quick reminder that the authors have posted their responses. The discussion period ends on December 2, so please review the rebuttal and share any follow-up comments as soon as possible. Your timely input is greatly appreciated. Thanks.

Best,

Your ACs

---

### Meta-Review · Area_Chair_3WZx · 2026-01-12

**Summary:**

This paper proposes AdaGUI-R1, a mobile GUI agent that introduces difficulty-aware reasoning, dynamically modulating chain-of-thought depth based on estimated step difficulty. The method combines a self-supervised “think–action” data construction pipeline with Group Adaptive Policy Optimization (GAPO), which introduces a difficulty-conditioned thought reward and a Gaussian exploration reward. Experiments on multiple offline GUI benchmarks show reduced reasoning tokens and moderate gains in step-level accuracy. Reviewers generally found the work well engineered and empirically thorough, but raised substantial concerns about novelty, evaluation validity, and real-world relevance.

**Reviewer Concerns:**

The reviewers’ concerns fall into three main categories.

(1) Limited novelty relative to prior adaptive-reasoning work.
Multiple reviewers noted that the core idea of adapting reasoning depth to difficulty is well established in prior literature, including difficulty-aware CoT, adaptive computation, and token-budget–aware reasoning. While the authors argue that GUI automation is multimodal and thus different from text-only settings, reviewers were not convinced this difference alone constitutes a substantive research advance. Several components, such as self-supervised CoT bootstrapping, soft spatial rewards, and difficulty-conditioned, incentivesclosely resemble existing methods (e.g., STaR-style bootstrapping and prior RL-based grounding rewards), with novelty primarily arising from recombination rather than new principles.

(2) Evaluation misalignment with real-world GUI agent performance.
A major unresolved concern is the reliance on offline, step-level metrics as the primary evaluation signal. Reviewers repeatedly emphasized that recent GUI agent work prioritizes online, full-task success rates in interactive environments. Although the authors provided full-trajectory success rates in rebuttal, these numbers are substantially lower than competitive agents and indicate that the proposed method does not translate well to realistic online settings. The clarification around step-level vs. trajectory-level success resolves a metric ambiguity, but does not alleviate the concern that the method’s empirical gains may not reflect practical utility.

(3) Questionable generality and cost–benefit trade-off.
Reviewers also questioned whether the gains stem from difficulty-aware reasoning itself or from additional supervision and fine-tuning on curated trajectories. The approach still depends on existing human-annotated datasets, raising concerns about scalability and data leakage. While the rebuttal added FLOPs analysis and ablations, reviewers remained unconvinced that the added system complexity and training overhead are justified given the modest improvements and limited online effectiveness.

Overall, although the rebuttal addressed many clarification and ablation requests, it did not resolve the core concerns about novelty, evaluation validity, and real-world impact, which ultimately drive the rejection recommendation.

**Reviewer Scores:**

Likely no changes. The reviewers unanimously voted negatively on this submission.

---

### Decision · Program_Chairs · 2026-01-26

Reject